# Principle-Evolvable Scientific Discovery via Uncertainty Minimization

**Yingming Pu** [1 2 3]   **Tao Lin** [2]   **Hongyu Chen** [2]

## Abstract

Large Language Model (LLM)-based scientific agents have accelerated scientific discovery, yet they often suffer from significant inefficiencies due to adherence to fixed initial priors. Existing approaches predominantly operate within a static hypothesis space, which restricts the discovery of novel phenomena, resulting in computational waste when baseline theories fail. To address this, we propose shifting the focus from searching hypotheses to evolving the underlying scientific principles. We present PIEVO, a principle-evolvable framework that treats scientific discovery as Bayesian optimization over an expanding principle space. By integrating Information-Directed Hypothesis Selection via Gaussian Process and an anomaly-driven augmentation mechanism, PIEVO enables agents to autonomously refine their theoretical worldview. Evaluation across four benchmarks demonstrates that PIEVO (1) achieves an average solution quality of up to 90.81%~93.15%, representing a 29.7%~31.1% improvement over the state-of-the-art, (2) attains an 83.3% speedup in convergence step via significantly reduced sample complexity by optimizing the compact principle space, and (3) maintains robust performance across diverse scientific domains and LLM backbones. Code is publicly available at github.com/amair-lab/PiEvo.

## 1. Introduction

Large Language Model (LLM)-based scientific agents have significantly accelerated the pace of automated scientific discovery (Gridach et al., 2025; Wei et al., 2025; Zhang et al., 2025; Tang et al., 2026; Lu et al., 2026; Yamada et al., 2025). By automating complex reasoning and hy-

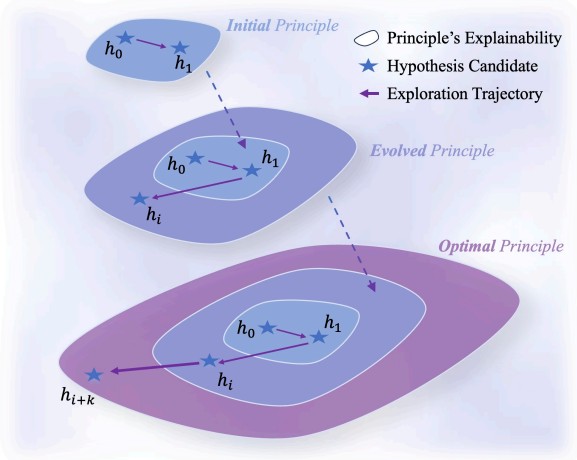

*Figure 1.* **Illustration of principle evolution.** PIEVO evolves the optimizable principle space by exploring hypothesis candidates progressively, and the system seeks a principle capable of explaining or predicting the observed phenomena (e.g., empirical observations).

pothesis generation, these systems have shown potential to transform research methodologies across fundamental fields, including chemistry (Boiko et al., 2023), biology (Mitchener et al., 2025), physics (Ghafarollahi & Buehler, 2025), and material science (Panapitiya et al., 2025). Despite these advancements, existing approaches predominantly maximize exploitation within a fixed search space.

While recent frameworks have begun to incorporate scientific principles as guidance (Pu et al., 2025; Lai & Pu, 2025; Zhao et al., 2025), they typically treat these principles, or the prior knowledge, as immutable constraints. Consequently, when agents encounter experimental evidence that contradicts their prior, they often discard valid observations as errors rather than recognizing them as signals for theoretical refinement and extension. This rigidity leads to significant inefficiencies: *agents explore the hypothesis space constrained by fixed priors without updating the "cognition" of real-world knowledge to emerging experimental evidence.*

These limitations culminate in three primary challenges: (a) **restricted scientific insight discovery**, where agents are blinded to novel phenomena that lie outside their initial prior; (b) **inefficiency in hypothesizing**, caused by the repetitive application of limited and inadequate prior knowledge; and (c) **inability to handle anomalies in discovery pro-**

[1] Zhejiang University, Hangzhou, Zhejiang, China. [2] Westlake University, Hangzhou, Zhejiang, China. [3] Email to: puyingming@westlake.edu.cn. Correspondence to: Tao Lin <lintao@westlake.edu.cn>.

*Proceedings of the 43rd International Conference on Machine Learning*, Seoul, South Korea. PMLR 306, 2026. Copyright 2026 by the author(s).

**cess**, where the system fails to leverage unexpected results to expand its conceptual boundaries.

To bridge this gap, we propose **PIEVO**, which reformulates scientific discovery as *Bayesian optimization over an evolving principle space*. Departing from the standard paradigm of static hypothesis searching (Pu et al., 2025; Tang et al., 2026), **PIEVO** models scientific principles (Definition 1.1) as learnable probabilistic priors that actively guide exploration. A principle is higher-level than a hypothesis: it constrains a family of plausible hypotheses rather than specifying a single testable candidate. Crucially, **PIEVO** employs an *anomaly-driven epistemic evolution* mechanism (Figure 1): high-surprisal experimental outcomes—typically discarded as noise—are leveraged to catalyze *principle space expansion* and belief updates. This fosters a *dual-loop optimization* cycle: refined principles constrain the search to high-utility manifolds, while anomalies drive the structural evolution of the principles themselves, ensuring the agent's internal model scales in complexity with empirical data.

**Definition 1.1** (Scientific Principles). *In this paper, we use the term* principle *to denote a high-level scientific mechanism that constrains which hypotheses are plausible, such as a textual statement describing a correlation, causal relation, or structural rule in a domain. See Example 3.1 for an instantiated principle-hypothesis pair.*

We evaluate **PIEVO** across four diverse benchmarks in physics, chemistry, biology, and materials science. Empirical results demonstrate that **PIEVO**: ❶ **Establishes state-of-the-art performance**, achieving an average solution quality of up to 93.15% and surpassing top-tier baseline (Pu et al., 2025) by up to 31.06%; ❷ **Enhances efficiency**, attaining 83.3% higher sample efficiency than PiFlow while reducing inference time by ∼16%; ❸ **Exhibits robust transferability** across tasks and LLMs; and ❹ **Uncovers fundamental mechanisms**, identifying novel principles in sub-wavelength chiral optics.

In summary, our contributions are:

(a) We introduce the concept of **Principle Evolution**, a paradigm shift that transforms scientific discovery from searching in a static hypothesis space to optimizing the underlying scientific principle space, which constrains vast regions of potential hypotheses.

(b) We propose **PIEVO**, a framework that automatically evolves tentative principles via Bayesian updates and adaptively extends the principle space based on experimental feedback, turning anomalies into drivers of scientific discovery.

(c) We conduct **extensive evaluations** across four real-world tasks, demonstrating that PIEVO improves discovery efficiency with 20%∼30% higher performance compared to SOTA, while validating its ability to aid the discovery of physical principles in a blind case study.

## 2. Related Work

### 2.1. Autonomous Agents for Scientific Discovery

The integration of Large Language Models (LLMs) into scientific workflows has catalyzed a paradigm shift from manual experimentation to autonomous discovery (Gridach et al., 2025; Wei et al., 2025; Zheng et al., 2025). Initial efforts focused on specialized assistants for specific domains, such as organic synthesis (M. Bran et al., 2024) and material design (Panapitiya et al., 2025). Recent advancements have expanded into general-purpose *AI Scientists* capable of end-to-end research management. Systems like The-AI-Scientist (Lu et al., 2026; Yamada et al., 2025), AI-Researcher (Tang et al., 2026), and Agent Laboratory (Schmidgall et al., 2025) demonstrate proficiency in ideation, coding, and manuscript generation. Concurrently, multi-agent frameworks have emerged to enhance reliability through collaborative reasoning (Su et al., 2025; Zhang et al., 2025) and comprehensive workflow management (Chai et al., 2025; Mitchener et al., 2025). However, a distinct limitation persists: these systems largely operate within the bounds of their pre-trained knowledge or static external retrievals. While principle-aware frameworks (Pu et al., 2025) introduce scientific rules as constraints, they treat these principles as immutable priors. This rigidity prevents agents from challenging foundational assumptions, restricting their ability to discover phenomena that contradict established theories (Wang et al., 2023; Xie et al., 2025).

### 2.2. Epistemic Evolution and Active Learning

To transcend static capabilities, recent research focuses on *self-evolving* agents that refine their behavior through interaction (Zhang et al., 2025; Zheng et al., 2025). Frameworks like RAGEN (Wang et al., 2025), Agent0 (Xia et al., 2025a), and EvolveR (Wu et al., 2025b) utilize reinforcement learning and self-generated curricula to optimize procedural strategies from experience. Parallel to this, Bayesian inference and Active Learning (AL) serve as the theoretical backbone for data-efficient decision-making in uncertainty-laden environments (Zhou et al., 2025; Xia et al., 2025b). These methodologies have been successfully adapted for LLM-driven planning (He et al., 2024) and curiosity-driven world modeling (Levy et al., 2025), enabling agents to maximize information gain with limited feedback. Despite these successes, a critical gap remains in the *target* of optimization. Existing evolutionary and Bayesian frameworks predominantly optimize *procedural parameters* (e.g., policy weights, prompts) or *fact accumulation*. They do not apply Bayesian updates to the *semantic principles* themselves. PIEVO bridges this gap by formalizing scientific principles not as fixed constraints, but as probabilistic beliefs to be iteratively optimized, thereby enabling the discovery of new physical laws through anomaly-driven active learning.

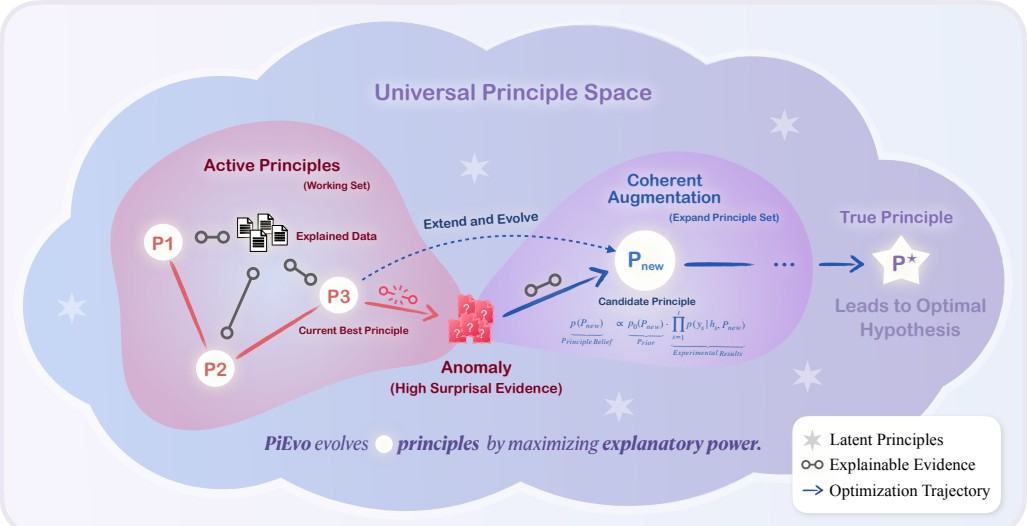

*Figure 2.* **Evolutionary trajectory of the principle space via Coherent Augmentation.** The system guided by PIEVO operates within a restricted *Active Principles*; however, when *high-surprisal* anomalies suggest epistemic stagnation, the system triggers a discovery phase. PIEVO integrates *new candidate principles* ($P_{new}$) to reconcile these anomalies while maintaining consistency with the whole historical evidence through *Coherent Augmentation*. This iterative evolution drives the optimization trajectory toward the True Principle $P^\star$, enabling the identification of the optimal hypothesis candidate.

## 3. Methodology

### 3.1. Overview

Figure 2 illustrates the general concept of *principle-evolving*. Unlike existing methods that operate with fixed hypothesis space (Tang et al., 2026; Lu et al., 2026; Yamada et al., 2025; Pu et al., 2025), PIEVO actively optimizes *scientific principles*, i.e., the natural-language proposition encoding a candidate scientific regularity, mechanism, or correlation. Leveraging Bayesian inference, it updates the posterior over principles using experimental evidence and extends its Active Principles to efficiently search hypotheses.

As illustrated in Figure 3, PIEVO is a strategic layer that connects to a minimal scientific agent system through prompt injection per iteration: (a) Principle Agent is guided to expand the Active Principles scope (see Section 3.4) or wait for other two agents to collect enough evidence, (b) Hypothesis Agent is guided by current Maximum A Posteriori (MAP) principle as a conditional prompt to propose hypothesis candidates (see Appendix K), and (c) Experiment Agent is guided to test the only one candidate selected by PIEVO (see Section 3.3). These modules establish the iteration-level directing, allowing to explore hypotheses from evolvable principle scope. We illustrate this principle-to-hypothesis-to-evidence pipeline with a nanohelix design example in Example 3.1.

**Example 3.1.** *A candidate **scientific principle** (P) states that "Maximum g-factor arises from Toroidal Anapole Resonance, which requires a tight helix radius, high turn density,*

*and a short pitch length to enable destructive multipolar interference." A **hypothesis** h is proposed: "A nanohelix with fiber radius of 20.0 nm, helix radius of 60.0 nm, 6.0 turns, pitch length of 20.0 nm, in the wavelength of 400.0 nm through the light source of Z axis and forward light direction, is expected to yield a high circular dichroism and a high g-factor response." **External validation function** $f^\star$ is the true underlying physical mapping, represented in our benchmarks by a high-fidelity surrogate model. It takes the concrete hypothesis h (i.e., structure parameters) as input and outputs the scalar observation y.*

### 3.2. Theoretical Framework

**Definition 3.2** (**Universal Principle Space**). *Let $\bar{\mathcal{P}}$ be a countable, universal set of potential scientific principles, endowed with a fixed prior distribution $p_0$ with finite entropy $H(p_0)$. At any time step $t$, the agent maintains a finite Active Principle $\mathcal{P}_t \subseteq \bar{\mathcal{P}}$. The goal is to identify the unknown True Principle $P^\star$ while maximizing the outcome of hypotheses generated under it.*

In this section, we first define the interaction between internal PIEVO optimization and the external physical validation, and then formulate the dual uncertainty minimization architecture in PIEVO, i.e., **Principle** $\xrightarrow{(a)}$ **Hypothesis** $\xrightarrow{(b)}$ **Evidence**.

**Hypothesis & Principle Space.** Let $\mathcal{H}$ denote the space of all possible experimental hypotheses (e.g., molecular structures) and $\bar{\mathcal{P}}$ denote the *universal set of scientific principles* (Definition 3.2). At any time $t$, the system maintains a set

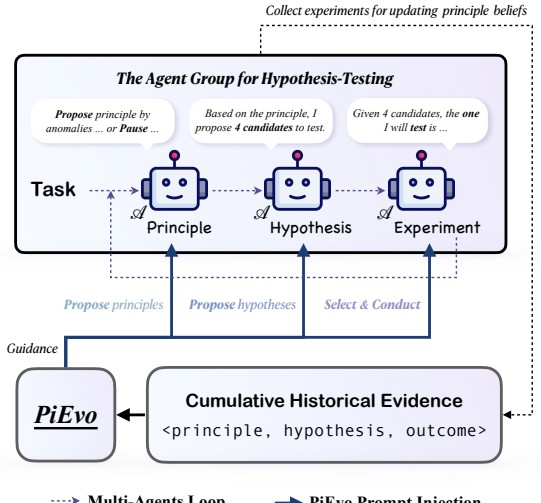

*Figure 3.* **Guidance Injection Mechanism.** PIEVO generates strategic guidance with a prompt template (see Appendix K) via backend optimization: (a) Active Principles, (b) identified high-surprisal anomalies and (c) hypothesis selection, injecting it into agent's context to guide reasoning.

of *Active Principles* $\mathcal{P}_t = \{P_1, \ldots, P_n\} \subset \bar{\mathcal{P}}$.

**External Validation Function** $f^\star$. An experiment consists of selecting a hypothesis $h \in \mathcal{H}$ and observing a real-valued outcome $y \in \mathbb{R}$ (e.g., bio-activity of a molecule). We model this as $y = f^\star(h) + \epsilon$, where $f^\star$ is the true underlying physical mapping (e.g., wet experiment, simulation, etc.) and $\epsilon$ is observational noise. In our study, $f^\star$ corresponds to surrogate model evaluations per task.

**Internal Principle Beliefs and Likelihoods.** The epistemic state of PIEVO in principle optimization is formalized as a *belief* distribution $p_t(P)$ for each principle $P \in \mathcal{P}_t$, updated via Bayesian inference given the interaction history $H_{t-1} = \{(h_s, y_s)\}_{s=1}^{t-1}$. To enable belief update, we model the *likelihood* $p(y \mid h, P)$ of observing an outcome $y$ given a hypothesis $h$ and a governing principle $P$ as a generative distribution, describing the expected experimental outcomes if $P$ were the true governing principle. With this *likelihood*, PIEVO can perform both hypothesis selection (see Section 3.3) and coherent principle augmentation (see Section 3.4).

To navigate $\bar{\mathcal{P}}$, PIEVO models two distinct uncertainties that obstruct the identification of $P^\star$ as a closed loop:

(a) **Principle→Hypothesis Uncertainty.** Given a principle as hypothesizing direction, this term captures the stochasticity in translating a high-level principle into a concrete, testable hypothesis $h$. In our study, this term is implemented via a well-instructed LLM to propose $h$ conditioned on one dynamically optimized principle.

(b) **Evidence→Principle Uncertainty.** Based on agent's history, this term represents the epistemic uncertainty

regarding which principle in the current Active Principle scope $\mathcal{P}_t$ is the gold choice. This term is grounded by hypothesis selection and principles optimization.

> **Overview of theoretical framework.** We formulate scientific discovery as a dual uncertainty problem over *hypothesis* and *principle* space, respectively:
> **(a) For *hypothesis* space**, PIEVO selects possible high-quality hypothesis to reduce the uncertainty of principle beliefs (see Section 3.3).
> **(b) For *principle* space**, PIEVO acts as a monitor to maintain a posterior distribution over the working principles, iteratively refining a set of principle-conditioned likelihood models, i.e., Gaussian Process Experts, to converge to the true principle (see Section 3.4).

### 3.3. Information-Directed Hypothesis Selection

Scientific discovery inherently requires balancing ***exploitation*** with ***exploration***. We illustrate this necessity through a motivating case in superconductor discovery:

**Example 3.3** (**Reward Trap of Paradigm Exploitation**). *Consider the search for room-temperature superconductors: Outcome-driven agents (e.g., PiFlow) rigorously hill-climb within established systems like* Cuprate *manifolds. While capable of local optimization, they remain blind to counter-intuitive regimes (e.g.,* Iron-based *families) that appear suboptimal under the current paradigm.*

To address this gap, we introduce hypothesis selection, while Hypothesis Agent is only responsible for suggesting diverse candidates and Experiment Agent for executing the one selected by PIEVO. By valuing *information gain* alongside *yield*, the system validates emerging principles that eventually unlock superior hypothesis regions.

**Information-Directed Hypothesis Selection.** By deducing from our decomposition of the scientific discovery problem, we adopt the Information-Directed Sampling (IDS) (Russo & Van Roy, 2014) to select the next hypothesis $h_t$ (details refer to Appendix ). IDS minimizes the objective:

$$h_t = \arg\min_{h \in \mathcal{H}} \left(\Delta_t(h)\right)^2 / I_t(h), \tag{1}$$

where $\Delta_t(h) = \mathbb{E}_{p_t(P)}[v^\star(P)] - \mathbb{E}_{p_t(P)}[\mu_P(h)]$ is the expected regret relative to the estimated optimal value $v^\star(P)$ under current beliefs. The denominator $I_t(h) = I(P; Y \mid h, H_{t-1})$ is the *Information Gain*, defined as the expected reduction in the entropy of the posterior $p_t(P)$. This ratio ensures that the agent only incurs high regret (exploration costs) if the expected information gain regarding the principle $P$ is commensurately high.

**Modeling Likelihoods by GP Experts.** To compute the required likelihoods $p(y \mid h, P)$ in a sample-efficient

manner, we utilize distinct *Gaussian Process (GP) experts* $\{\mathcal{M}_P\}_{P \in \mathcal{P}_t}$ rather than neural alternatives for three reasons. (i) **Data efficiency**: GPs provide robust generalization and exact Bayesian inference in *sparse-data regimes* where experimental budgets are severely constrained (e.g., $T < 30$). (ii) **Calibrated uncertainty**: Unlike typical deep models, GPs yield properly calibrated predictive variances $\sigma_P^2(h)$, which are essential for the information-theoretic selection in Equation 1. (iii) **Analytical computation**: GPs allow for the analytical computation of $\Delta_t(h)$ and $I_t(h)$ via the BALD approximation (Houlsby et al., 2011), bypassing the need for computationally expensive ensemble retraining.

To bridge the modality gap between textual propositions and numerical GP kernels, we employ semantic alignment as a theoretically rigorous proxy for physical likelihoods (proof in Appendix B):

$$\phi(h, P) = \begin{bmatrix} e_h \cdot e_P, & \|e_h - e_P\|_2 \end{bmatrix}^\top, \qquad (2)$$

where $e_h$ and $e_P$ are normalized embeddings of the hypothesis and principle, respectively. This allows us to leverage light efforts to achieve likelihood estimation, enabling efficient GPs training and inference.

> **Takeaway (Hypothesis Selection in PIEVO).** Overall, we use GPs to model likelihoods $p(y|h, P)$ for updating the principle beliefs and selecting the exploration-exploitation balanced hypothesis for testing.

### 3.4. Coherent Augmentation for Principles

In scientific discovery, the initial conception of the external world is often incomplete. If the true principle is missing, the agent may fall into epistemic stagnation. PIEVO resolves this through an anomaly-driven expansion of the principle space, including (a) identifying anomalies and (b) reconciling anomalies by proposing a new principle, as detailed below.

**Identifying Anomalies.** The system monitors the validity of principles by first computing the *Anomaly Score* $S_s$, a bounded metric proxying the epistemic surprise of experimental evidence. This score quantifies the degree of mismatch between observation and GP prediction, accounting for both model uncertainty and observation noise. For a new observation $(h_s, y_s)$, the *Anomaly Score* under the current Maximum A Posteriori (MAP) principle $P_t^{\mathrm{MAP}}$ is:

$$S_s = 1 - \exp\left(-\sqrt{\frac{(y_s - \mu_{\mathrm{MAP}}(h_s))^2}{\sigma_{\mathrm{MAP}}^2(h_s) + \sigma_{\mathrm{obs}}^2}}\right) > \theta_t, \quad (3)$$

where $\theta_t \in (0, 1)$ is anomaly threshold, $\mu_{\mathrm{MAP}}(h_s)$ and $\sigma_{\mathrm{MAP}}^2(h_s)$ are the predictive mean and variance under the MAP principle, and $\sigma_{\mathrm{obs}}^2$ is the observation noise variance.

An observation is flagged as anomalous if $S_s > \theta_t$. A collection of high-surprisal hypotheses forms the *Anomaly Set* $\mathcal{U}_t$, signaling that the current principle space $\mathcal{P}_t$ is insufficient to explain the observed phenomena.

**Reconciling Anomaly with Coherent Principle Augmentation.** Upon detected anomalies, PIEVO triggers a discovery phase where the Principle Agent proposes a new candidate $P_{\mathrm{new}} \in \bar{\mathcal{P}}$ to resolve the contradictions in $\mathcal{U}_t$. Finally, the posterior belief of Active Principles $\mathcal{P}_t$ is re-calculated over the augmented set $\mathcal{P}_{t+1} = \mathcal{P}_t \cup \{P_{\mathrm{new}}\}$ by:

$$p_{t+1}(P) \propto p_0(P) \prod_{s=1}^{t} p(y_s \mid h_s, P). \qquad (4)$$

The new principle is only adopted if it provides superior explanatory power for $H_t$.

**Theorem 3.4** (Convergence and Efficiency, Informal). *Under coherent principle augmentation and a calibrated generator, the* PIEVO *optimization loop guarantees (formally stated in Theorem A.6): (a) **sublinear cumulative regret**, where the cumulative regret follows $R(T) = \tilde{\mathcal{O}}(\sqrt{T})$, and (b) **posterior consistency**, where the system belief $p_t(P)$ concentrates on the optimal principle $P^\star$.*

In summary, we detailed the implementation of PIEVO in Algorithm 1 with prompt templates in Appendix K, and the optimization in PIEVO provides strong theoretical guarantees (Theorem 3.4, with proof in Appendix A). Additionally, its superior efficiency demonstrates faster convergence than PiFlow (Pu et al., 2025) (see proof in Appendix C), as also empirically validated in Figure 7.

> **Takeaway (Coherent Augmentation in PIEVO).** The surprisal-based anomaly detection allows both sensitivity control by threshold and dynamic principle reconciling. This self-closed updating ensures that the scientific cognition evolves coherently with empirical evidence.

## 4. Experiments

### 4.1. Experiment Setup

**Tasks and Benchmarks.** We evaluate PIEVO on four standard scientific discovery benchmarks: nanomaterial optical property optimization (NHO), molecular bio-activity optimization (MBO), and superconductor critical temperature optimization (SPO) (Pu et al., 2025), covering **continuous** (seven dimension), **discrete**, and **hybrid** search spaces, alongside transition metal complex optimization (TMC) (Song et al., 2025) for four-dimensional **combinational** entities search. Employing high-fidelity surrogate models, each task aims to maximize the target property value within a constrained experiment budget (See Appendix J).

*Table 1.* **Performance comparison with baselines on 4 benchmarks under the Solution Quality (SQ %) metric.** We categorize the results by LLMs, i.e., `Qwen3-32B` and `Gemini-2.5-Flash` with both non-thinking mode, and highlight superior or inferior SQ compare to Vanilla MAS. Our PIEVO outperforms all baselines, and achieves $1.5 \sim 1.7$ times SQ of Vanilla MAS in average.

| Model / Method | MBO | NHO | SPO | TMC | Average |
|---|---|---|---|---|---|
| **Qwen3-32B** (No Thinking) | | | | | |
| Vanilla MAS | $57.57 \pm 3.50$ | $58.22 \pm 8.35$ | $28.71 \pm 3.20$ | $63.98 \pm 4.79$ | $52.12$ |
| ReAct (Yao et al., 2023) | $69.15 \pm 8.88$ ↑1.2x | $67.02 \pm 3.81$ ↑1.2x | $30.51 \pm 0.05$ ↑1.1x | $58.07 \pm 5.08$ ↓0.9x | $56.19$ ↑1.1x |
| The AI Scientist v1 (Lu et al., 2026) | $66.41 \pm 12.48$ ↑1.2x | $79.76 \pm 0.58$ ↑1.4x | $30.52 \pm 0.26$ ↑1.1x | $72.36 \pm 8.72$ ↑1.1x | $62.26$ ↑1.2x |
| The AI Scientist v2 (Yamada et al., 2025) | $76.56 \pm 2.04$ ↑1.3x | $71.50 \pm 2.97$ ↑1.2x | $30.01 \pm 0.76$ ↑1.0x | $71.45 \pm 1.59$ ↑1.1x | $62.38$ ↑1.2x |
| AI Researcher (Tang et al., 2026) | $69.72 \pm 11.46$ ↑1.2x | $79.05 \pm 4.28$ ↑1.4x | $30.24 \pm 1.21$ ↑1.1x | $74.17 \pm 3.27$ ↑1.2x | $63.29$ ↑1.2x |
| PiFlow (Pu et al., 2025) | $96.10 \pm 4.85$ ↑1.7x | $79.68 \pm 0.97$ ↑1.4x | $33.99 \pm 2.65$ ↑1.2x | $70.35 \pm 10.03$ ↑1.1x | $70.03$ ↑1.3x |
| **PIEVO (ours)** | $\mathbf{149.06} \pm 16.02$ ↑2.6x | $\mathbf{96.36} \pm 1.70$ ↑1.7x | $\mathbf{37.33} \pm 1.36$ ↑1.3x | $\mathbf{80.49} \pm 1.93$ ↑1.3x | $\mathbf{90.81}$ ↑1.7x |
| **Gemini-2.5-Flash-No Thinking** | | | | | |
| Vanilla MAS | $72.72 \pm 17.45$ | $63.77 \pm 22.22$ | $32.57 \pm 3.51$ | $79.36 \pm 4.86$ | $62.10$ |
| ReAct (Yao et al., 2023) | $61.24 \pm 14.71$ ↓0.8x | $73.70 \pm 9.63$ ↑1.2x | $32.57 \pm 3.46$ | $78.71 \pm 10.60$ ↓1.0x | $61.56$ ↓1.0x |
| The AI Scientist v1 (Lu et al., 2026) | $50.37 \pm 7.97$ ↓0.7x | $77.40 \pm 1.26$ ↑1.2x | $36.57 \pm 0.01$ ↑1.1x | $84.14 \pm 2.19$ ↑1.1x | $62.12$ |
| The AI Scientist v2 (Yamada et al., 2025) | $59.83 \pm 8.64$ ↓0.8x | $77.85 \pm 4.44$ ↑1.2x | $36.74 \pm 0.07$ ↑1.1x | $84.86 \pm 0.65$ ↑1.1x | $64.82$ ↑1.0x |
| AI Researcher (Tang et al., 2026) | $61.22 \pm 21.80$ ↓0.8x | $72.46 \pm 15.13$ ↑1.1x | $36.95 \pm 0.28$ ↑1.1x | $83.79 \pm 2.50$ ↑1.1x | $63.61$ ↑1.0x |
| PiFlow (Pu et al., 2025) | $80.18 \pm 7.55$ ↑1.1x | $79.32 \pm 4.60$ ↑1.2x | $37.31 \pm 0.71$ ↑1.1x | $87.45 \pm 1.54$ ↑1.1x | $71.07$ ↑1.1x |
| **PIEVO (ours)** | $\mathbf{153.53} \pm 6.35$ ↑2.1x | $\mathbf{87.98} \pm 1.13$ ↑1.4x | $\mathbf{37.85} \pm 0.35$ ↑1.2x | $\mathbf{93.25} \pm 0.79$ ↑1.2x | $\mathbf{93.15}$ ↑1.5x |

*Figure 4.* **Trajectory comparison among baselines w/ Qwen3-32B.** We plot the cumulative best solution quality over time for each task. SQ > 100% indicates surpassing the empirical reference. Our PIEVO consistently outperforms baselines across all tasks, though in challenging tasks like TMC and SPO, the gap is smaller due to the inherent searching complexity of hypotheses.

**Baselines.** We benchmark PIEVO against three categories of Multi-Agent Systems (MAS): (a) **Classical Strategies**: including *Vanilla MAS*, an iterative hypothesis-testing loop without higher-order strategy, and *ReAct* (Yao et al., 2023), which interleaves reasoning traces with actions. (b) **Autonomous Scientific Agents**: tailored adaptations of *AI-Researcher* (Tang et al., 2026) and *The-AI-Scientist (v1/v2)* (Lu et al., 2026; Yamada et al., 2025), restricted to their core hypothesis-generation modules without web searching and retrieval for fair comparison. (c) **Principle-Guided Approaches**: represented by *PiFlow* (Pu et al., 2025), the state-of-the-art method conditioning hypothesis generation on static scientific principles. Other methods like Co-scientist (Gottweis et al., 2025), Kosmos (Mitchener et al., 2025) and Robin (Ghareeb et al., 2025) are excluded from this comparative analysis, as they are either tailored to specific scientific domains or leverage hypothesis-testing mechanisms represented by the selected baselines.

**Implementation details.** For all experiments, we enforce a strict budget of 24 trials and truncate agent context to the 10 most recent messages to mitigate overflow. Performance metrics are reported as the mean $\pm$ std over three independent runs. We use `Qwen3-32B` (Yang et al., 2025) with `no_think` prompt for force-disabling Chain-of-Thought and `Gemini-2.5-Flash-No Thinking` backends with a temperature of 0.6. An additional `think` mode ablation study uses `Gemini-2.5-Flash-Thinking` (Comanici et al., 2025) through API.

### 4.2. Evaluation Metrics

To comprehensively evaluate the performance, we employ multi-faceted metrics that capture abilities to (a) reaching the objective, (b) exploration width and (c) promising exploitation, as detailed below.

**a. Solution Quality (SQ).** Defined as the maximum outcome achieved within its hypothesis-outcome trajectory $\mathcal{T}$, and outcome is normalized by the theoretical (NHO and SPO) or empirical (MBO and TMC) maximum, $\mu_{ref}$ (see Appendix J) per task:

$$\text{SQ} = {}^{\max\{y_k|(h_k,y_k)\in\mathcal{T}\}}\!/_{\mu_{ref}} \times 100\%.$$

A high SQ exceeds 100% indicates discovery of solutions superior to the reference dataset/maximum.

*Table 2.* **Exploration (APD ↑) and Exploitation (AUOC ↑) comparison with baselines on 4 benchmarks.** We categorize the results by LLMs, i.e., `Qwen3-32B` and `Gemini-2.5-Flash` with both non-thinking mode, and highlight superior or inferior average performance compare to Vanilla MAS. Our PIEVO attains superior balance in both exploration and exploitation, as shown in Figure 5.

| Model / Method | MBO | | NHO | | SPO | | TMC | | Average | |
|---|---|---|---|---|---|---|---|---|---|---|
| | APD ↑ | AUOC ↑ | APD ↑ | AUOC ↑ | APD ↑ | AUOC ↑ | APD ↑ | AUOC ↑ | Avg APD ↑ | Avg AUOC ↑ |
| **Gemini-2.5-Flash-No Thinking** | | | | | | | | | | |
| Vanilla | $6.0 \pm 1.8$ | $61.6 \pm 19.6$ | $34.5 \pm 46.8$ | $59.6 \pm 20.8$ | $7.0 \pm 8.9$ | $32.2 \pm 2.9$ | $44.9 \pm 29.0$ | $76.7 \pm 2.7$ | 23.1 | 57.5 |
| ReAct | $7.3 \pm 1.7$ | $54.9 \pm 12.7$ | $11.2 \pm 8.5$ | $69.3 \pm 12.7$ | $2.6 \pm 4.3$ | $32.4 \pm 3.2$ | $45.2 \pm 8.4$ | $77.0 \pm 9.1$ | $16.6^{\downarrow 0.7x}$ | $58.4^{\uparrow 1.0x}$ |
| The AI Scientist v1 | $3.3 \pm 0.6$ | $43.5 \pm 6.7$ | $14.2 \pm 4.0$ | $76.5 \pm 1.3$ | $3.2 \pm 0.2$ | $34.5 \pm 3.0$ | $38.9 \pm 3.8$ | $83.8 \pm 2.5$ | $14.9^{\downarrow 0.6x}$ | $59.6^{\uparrow 1.0x}$ |
| The AI Scientist v2 | $6.5 \pm 2.6$ | $50.1 \pm 7.1$ | $27.2 \pm 1.7$ | $77.2 \pm 4.3$ | $4.1 \pm 0.3$ | $36.2 \pm 0.1$ | $52.7 \pm 3.9$ | $80.7 \pm 2.4$ | $22.6^{\downarrow 1.0x}$ | $61.1^{\uparrow 1.1x}$ |
| AI Researcher | $6.9 \pm 8.2$ | $49.1 \pm 10.0$ | $34.1 \pm 6.9$ | $60.4 \pm 23.2$ | $7.6 \pm 5.0$ | $36.1 \pm 0.3$ | $55.5 \pm 11.6$ | $50.3 \pm 25.7$ | $26.0^{\uparrow 1.1x}$ | $49.0^{\downarrow 0.9x}$ |
| PiFlow | $14.4 \pm 8.9$ | $52.8 \pm 6.1$ | $36.5 \pm 8.7$ | $66.0 \pm 6.7$ | $16.2 \pm 6.4$ | $33.4 \pm 4.1$ | $61.1 \pm 11.4$ | $80.9 \pm 5.5$ | $32.0^{\uparrow 1.4x}$ | $58.3^{\uparrow 1.0x}$ |
| **PIEVO (ours)** | $\mathbf{28.3} \pm \mathbf{2.5}$ | $\mathbf{122.3} \pm \mathbf{9.1}$ | $\mathbf{82.5} \pm \mathbf{24.7}$ | $\mathbf{85.9} \pm \mathbf{2.0}$ | $\mathbf{21.1} \pm \mathbf{2.5}$ | $\mathbf{35.4} \pm \mathbf{0.9}$ | $\mathbf{86.8} \pm \mathbf{4.0}$ | $\mathbf{92.3} \pm \mathbf{1.3}$ | $\mathbf{54.7}^{\uparrow 2.4x}$ | $\mathbf{84.0}^{\uparrow 1.5x}$ |
| **Qwen3-32B** | | | | | | | | | | |
| Vanilla | $9.6 \pm 1.6$ | $53.2 \pm 1.1$ | $9.3 \pm 1.7$ | $57.8 \pm 7.8$ | $5.8 \pm 5.0$ | $28.6 \pm 3.4$ | $34.3 \pm 22.3$ | $62.4 \pm 3.3$ | 14.8 | 50.5 |
| ReAct | $3.1 \pm 2.0$ | $63.7 \pm 7.2$ | $6.9 \pm 8.8$ | $67.0 \pm 3.8$ | $1.1 \pm 1.2$ | $30.5 \pm 0.0$ | $47.2 \pm 21.0$ | $57.8 \pm 4.7$ | $14.6^{\downarrow 1.0x}$ | $54.8^{\uparrow 1.1x}$ |
| The AI Scientist v1 | $9.1 \pm 4.1$ | $53.2 \pm 5.9$ | $17.4 \pm 1.3$ | $71.4 \pm 3.4$ | $2.3 \pm 1.6$ | $30.3 \pm 0.4$ | $53.2 \pm 18.7$ | $69.8 \pm 6.3$ | $20.5^{\uparrow 1.4x}$ | $56.2^{\uparrow 1.1x}$ |
| The AI Scientist v2 | $7.2 \pm 1.5$ | $58.9 \pm 5.3$ | $22.4 \pm 2.1$ | $68.3 \pm 1.2$ | $4.4 \pm 0.4$ | $29.8 \pm 0.6$ | $62.5 \pm 2.6$ | $66.5 \pm 2.3$ | $24.1^{\uparrow 1.6x}$ | $55.9^{\uparrow 1.1x}$ |
| AI Researcher | $9.0 \pm 7.5$ | $53.0 \pm 10.8$ | $38.9 \pm 14.9$ | $74.9 \pm 5.9$ | $7.3 \pm 4.0$ | $29.8 \pm 0.9$ | $44.2 \pm 4.7$ | $68.9 \pm 4.3$ | $24.9^{\uparrow 1.7x}$ | $56.6^{\uparrow 1.1x}$ |
| PiFlow | $19.0 \pm 14.5$ | $80.4 \pm 17.5$ | $59.2 \pm 20.1$ | $70.3 \pm 4.8$ | $\mathbf{35.2} \pm \mathbf{21.7}$ | $32.1 \pm 3.6$ | $65.9 \pm 24.6$ | $68.7 \pm 8.8$ | $44.8^{\uparrow 3.0x}$ | $62.9^{\uparrow 1.2x}$ |
| **PIEVO (ours)** | $\mathbf{24.5} \pm \mathbf{2.6}$ | $\mathbf{118.3} \pm \mathbf{23.8}$ | $\mathbf{78.3} \pm \mathbf{31.6}$ | $\mathbf{95.7} \pm \mathbf{1.8}$ | $18.2 \pm 5.7$ | $\mathbf{33.2} \pm \mathbf{2.8}$ | $\mathbf{77.5} \pm \mathbf{7.0}$ | $\mathbf{69.7} \pm \mathbf{9.2}$ | $\mathbf{49.7}^{\uparrow 3.4x}$ | $\mathbf{79.2}^{\uparrow 1.6x}$ |

*Table 3.* **Performance on NHO task with different searching dimensions.** The best and runner-up results are **bolded** and underlined, respectively. Physical meaning of each dimension and performance interpretation see Appendix J.1.

| Method | dimension=4 | dimension | dimension=6 | Avg. |
|---|---|---|---|---|
| AI Researcher | 79.37 | 86.74 | 74.48 | 80.20 |
| The AI Scientist v1 | **82.47** | 83.81 | 81.01 | 82.43 |
| The AI Scientist v2 | 79.92 | 68.15 | 68.58 | 72.22 |
| PiFlow | 75.62 | 74.10 | 69.07 | 72.93 |
| **PIEVO (ours)** | 82.33 | **88.94** | **82.41** | **84.56** |

**b. Average Pairwise Distance (APD).** A value that describes average pairwise distance of all valid hypothesis, featurized by $\phi(h)$ per task (refer to Appendix J):

$$\text{APD} = {2}/{M(M-1)} \sum_{i<j}^{M} d\left(\phi(h_i), \phi(h_j)\right) .$$

A higher APD indicates searching broadly, which is a prerequisite for novel discovery and landscape coverage.

**c. Area Under the Optimization Curve (AUOC).** A value that measures the cumulative maximum performance over the fixed budget:

$$\text{AUOC} = {1}/{T \cdot \mu_{\text{ref}}} \sum_{t=1}^{T} \max_{k \leq t}\{y_k\} .$$

A higher AUOC signifies superior search efficiency that identifies promising directions early and exploits them effectively.

### 4.3. Performance Analysis

Table 1 presents a comparison of PIEVO against six baselines across four benchmarks, while Table 3 examines scala-

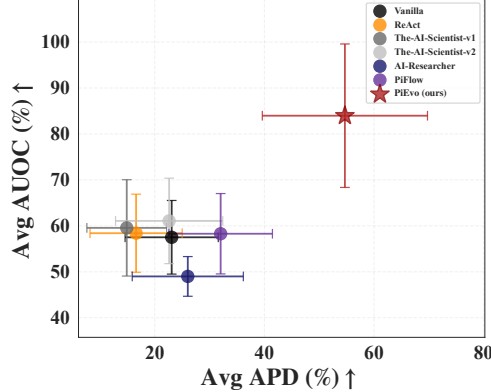

*Figure 5.* **Pareto frontier across all tasks.** PIEVO stands on the Pareto frontier, indicating its superior balance on both exploration (APD) and exploitation (AUOC) in scientific discovery.

bility under varying geometric dimensionality in the NHO task. Our analysis yields three primary observations:

**Obs.❶: PIEVO establishes a new state-of-the-art in solution quality.** PIEVO dominates all baselines, achieving an average SQ of $90.81\% \sim 93.15\%$. This corresponds to a **30% improvement** over leading AI Scientist frameworks ($62.26\% \sim 64.82\%$) and principle-conditioned PiFlow ($70.03\% \sim 71.07\%$), confirming that dynamic principle evolution effectively confines exploration to high-value scientific manifolds.

**Obs.❷: PIEVO effectively balances Exploration-Exploitation.** As visualized in the Pareto frontier (Figure 5), PIEVO demonstrates dominance in both diversity ($APD = 54.7\%$) and convergence efficiency ($AUOC = 84.0\%$). Unlike baselines that compromise diversity for hill-climbing speed (e.g., AI Researcher), PIEVO's dual-loop mechanism ensures broad coverage of

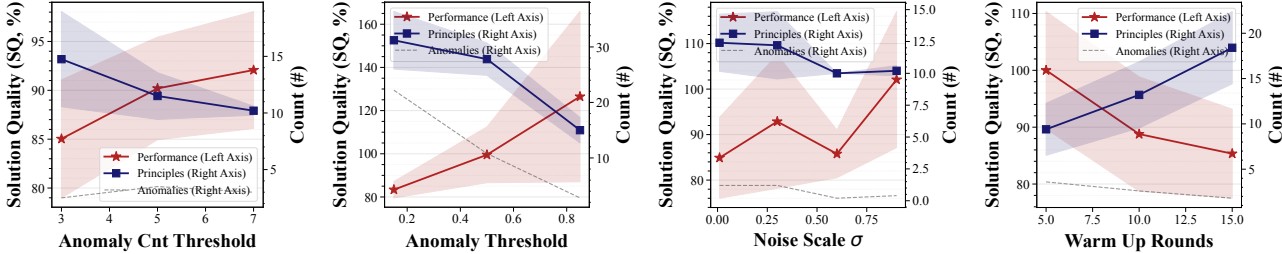

*Figure 6.* **Ablations of Parameters Sensitivity.** Both anomaly count threshold and anomaly threshold affect the number of principles, while sigma as noise representation affects the system's exploration capability. Warm-up rounds affect the system's initialization capability.

the hypothesis space without sacrificing local optimization.

**Obs.❸: PIEVO exhibits robustness in high-dimensional search spaces.** In the NHO task, where search space complexity scales with geometric dimensionality (Table 3), PIEVO sustains a high average SQ of $84.56\%$ across dimensions 4 to 6. Conversely, baselines suffer sharp performance collapse as dimensionality increases (e.g., PiFlow drops to $72.93\%$), demonstrating PIEVO's superior scalability in navigating complex design spaces.

**Obs.❹: PIEVO outperforms domain-agnostic optimizers.** We evaluate PIEVO against standard domain-agnostic optimizers (Table 4), including Random Search (RS), Genetic Algorithm (GA), Bayesian Optimization (BO), and Differential Evolution (DE). These solvers (BO, DE) struggle in few-shot regimes because they build response surfaces from scratch. Conversely, PIEVO acts as a zero-shot prior generator, alleviating cold start and constraining search toward high-yield manifolds.

*Table 4.* **Comparison against domain optimizers.** SQ % under the same 24 budget is reported. PIEVO's results can be found in Table 1.

| Method | MBO | NHO | SPO | TMC | Avg. |
|---|---|---|---|---|---|
| **RS** | $62.7 \pm 4.1$ | $39.5 \pm 16.3$ | $35.8 \pm 1.2$ | $70.0 \pm 5.4$ | 52.0 |
| **GA** | $65.9 \pm 7.2$ | $44.8 \pm 19.3$ | $39.4 \pm 2.6$ | $74.2 \pm 6.3$ | 56.1 |
| **BO** | $70.2 \pm 9.5$ | $64.8 \pm 12.2$ | $\mathbf{42.6 \pm 1.0}$ | $76.6 \pm 4.7$ | 63.5 |
| **DE** | $62.7 \pm 4.1$ | $51.2 \pm 14.3$ | $37.2 \pm 1.6$ | $70.0 \pm 5.4$ | 55.3 |
| **PIEVO w/ Qwen** | $149.1 \pm 16.0$ | $\mathbf{96.4 \pm 1.7}$ | $37.3 \pm 1.4$ | $80.5 \pm 1.9$ | 90.8 |
| **PIEVO w/ Gemini** | $\mathbf{153.5 \pm 6.4}$ | $88.0 \pm 1.1$ | $37.9 \pm 0.4$ | $\mathbf{93.3 \pm 0.8}$ | **93.2** |

### 4.4. Efficiency Analysis

We evaluate the discovery efficiency of PIEVO through two key dimensions: solution quality (SQ) and computational cost. As detailed in Table 5 and Figure 7, PIEVO demonstrates substantial gains in both metrics.

**Obs.❺: PIEVO achieves sublinear regret and accelerated convergence.** Empirical trajectories in Figure 7 validate the theoretical sublinear regret bound $\tilde{\mathcal{O}}(\sqrt{T})$ (Theorem 3.4). Specifically, PIEVO achieves an **83.3% speedup** in convergence step compared to PiFlow. Curve fitting reveals a significantly lower regret coefficient for PIEVO ($13.49\sqrt{T}$) versus PiFlow ($37.91\sqrt{T}$). This $\sim 3\times$ reduction empirically corroborates our theoretical insight: optimizing over the

*Table 5.* **Performance and efficiency comparison on NHO task, as SQ reported in Table 1.** The $\tau_{run}$ denotes the execution time in seconds. Efficiency is calculated as $\mathrm{SQ}/\log_{10}(\tau_{run})$.

| Method | SQ (%) ↑ | Time $\tau$ (sec) ↓ | $\frac{\mathrm{SQ}}{\log_{10}(\tau_{run})}$ ↑ |
|---|---|---|---|
| AI Researcher | $79.05 \pm 4.28$ | 3272.9 | 22.48 |
| The AI Scientist v1 | $79.76 \pm 0.58$ | 1953.5 | 24.23 |
| The AI Scientist v2 | $71.50 \pm 2.97$ | 1953.5 | 21.72 |
| PiFlow | $79.68 \pm 0.97$ | 2498.9 | 23.45 |
| **PIEVO (ours)** | $\mathbf{96.36 \pm 1.70}$ | 2096.7 | **29.01** |

principle variable (entropy $H(P)$) is more sample-efficient than searching the larger hypothesis space (scaling with $\log |\mathcal{H}|$), i.e., $\sqrt{H(P) \cdot T} < \sqrt{\log |\mathcal{H}| \cdot T}$.

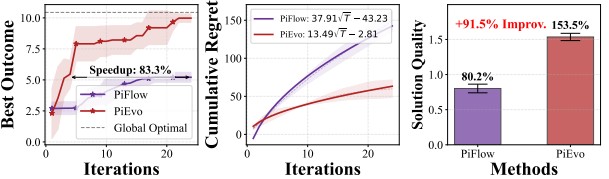

*Figure 7.* **Comparison of PiFlow and PIEVO in MBO task, as shown in Table 1 and Table 2.** PIEVO excels in 83.3% speedup and lower empirical regret compared to PiFlow, demonstrating its superior efficiency.

### 4.5. Case Study: Discovering Physics Mechanism

**Task Scenario.** We deploy PIEVO on a challenging discovery task in nanophotonics: identifying the electrodynamic origin of circular dichroism (CD) in *isolated* nanohelices. Unlike well-studied helix arrays (Faniayeu et al., 2020), the governing laws for single-structure chirality remain largely unexplored. PIEVO with agents are initialized without any document resources and must evolve principles to explain the chirality obtained from a high-fidelity surrogate model.

**Obs.❻: PIEVO autonomously identifies electrodynamic mechanisms.** Within just 30 iterations, PIEVO successfully identifies a high-surprisal anomaly. Triggered by this gap, PIEVO synthesizes a unified theory attributing strong chirality to the interference between *toroidal* and *electric quadrupole moments*. This mechanism was subsequently corroborated by full-wave FDTD analysis manually (see Appendix I), demonstrating the capacity of PIEVO to uncover physical laws in complex systems.

### 4.6. Ablation Study

**Hyperparameters Sensitivity.** We analyze PIEVO's sensitivity to key hyperparameters: anomaly threshold $\theta_t$ (Eq. 3), observational noise $\sigma$, and IDS warm-up rounds. Figure 6 demonstrates robustness across a wide $\theta_t$ spectrum, where the optimal value balances anomaly detection against noise rejection. Regarding warm-up, minimal initial exploration ($\approx 5$ rounds) proves sufficient for GP experts to establish meaningful priors, effectively minimizing uncertainty.

**PIEVO Module Ablations.** To disentangle component contributions, we perform ablations on the NHO task (Table 6, Figure 8). Disabling *Coherent Augmentation* (`Static Evolution`) caps SQ at $85.87\%$, as the agent cannot access high-value regimes beyond the initial prior. Similarly, removing *Information-Directed Selection* (`Greedy`) reduces SQ to $84.03\%$, limiting the efficient resolution of principle-evidence uncertainty. The full PIEVO framework achieves $96.36\%$ SQ, confirming that the synergy between dynamic principle evolution and information-directed sampling is critical for robust discovery.

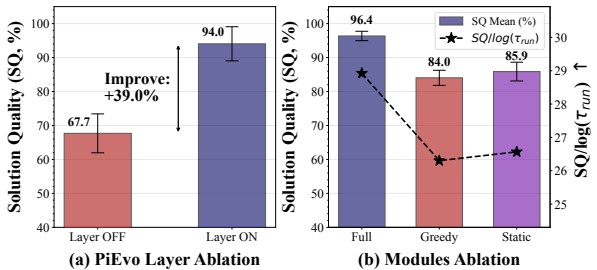

*Figure 8.* **Ablation study of PIEVO module on NHO task.** (a) Performance gain from PIEVO layer. (b) Comparison of cost-effectiveness across ON/OFF submodules in PIEVO.

*Table 6.* **Ablation study of PIEVO.** The mode `Greedy Only` is for disabling hypothesis selection, and `Static Evolution` is for disabling principle optimization in NHO task. Efficiency is calculated as SQ per log-scale time.

| Method | SQ (%) ↑ | SQ_max (%) ↑ | $\tau_{run}$ (sec) ↓ | $\frac{SQ_{max}}{\log_{10}(\tau_{run})}$ ↑ |
|---|---|---|---|---|
| **PIEVO** | **96.36** ± 1.70 | **99.64** | 2069.7 | **29.01** |
| w/o Hypothesis Selection | | | | |
| Greedy Only | 84.03 ± 2.22 | 86.99 | 2029.2 | 26.30 |
| w/o Principle Evolution | | | | |
| Static Evolution | 85.87 ± 2.76 | 88.63 | 2168.3 | 26.57 |

**Ablations of LLM Capability and GP Accuracy.** To validate low-resource robustness, we evaluated PIEVO using the lightweight Qwen3-14B (Table 7). It maintains strong performance and significantly outperforms baselines, proving it does not rely on computationally expensive reasoning models. Furthermore, geometric alignment is a robust proxy for experimental likelihood, GP experts associated with valid principles consistently achieve high accuracy across all tasks (Table 8). Crucially, a low $R^2$ is an intended epistemic signal, not a failure.

*Table 7.* **Performance comparison using lightweight Qwen3-14B backbone.** The methods are evaluated on the MBO task. The Drop% indicates the SQ% drop compared to PIEVO in this table.

| Method | SQ % | AUOC % | APD % | Drop % |
|---|---|---|---|---|
| PIEVO (ours) | **130.01** ± 52.60 | **108.3** ± 40.9 | **24.5** ± 13.7 | - |
| PiFlow | 59.99 ± 9.84 | 50.3 ± 10.7 | 9.7 ± 6.2 | - 53.86% |
| AI-Researcher | 49.42 ± 15.45 | 40.3 ± 2.9 | 3.3 ± 3.0 | - 61.99% |
| The-AI-Scientist-v2 | 61.87 ± 4.75 | 61.2 ± 5.6 | 2.4 ± 1.5 | - 52.41% |
| The-AI-Scientist-v1 | 67.09 ± 13.34 | 60.7 ± 9.8 | 4.4 ± 4.8 | - 48.40% |

*Table 8.* **Proportion of valid principles across tasks.** The validity of principles is determined by their $R^2$ correlation with solution quality, where a principle is considered valid if the correlation coefficient of GP's predictions and experiment results $R^2 > 0.1$, refer to Figure 12b for example.

| Task | Mean and std of $R^2$ | Valid Principles Proportion |
|---|---|---|
| MBO | 0.94 ± 0.06 | 80.0% |
| NHO | 0.81 ± 0.13 | 42.9% |
| SPO | 0.87 ± 0.08 | 95.0% |
| TMC | 0.81 ± 0.13 | 93.9% |

**Ablation of GP's Kernel.** We evaluate an Additive Kernel against our standard RBF kernel on this 2D space (Table 9). While the Additive Kernel shows a slightly higher mean SQ, it introduces severe instability (massive variance) and degrades exploitation-exploration balance (lower AUOC and APD). Thus, the RBF kernel (Algorithm 1) is the optimal choice for our low-dimensional feature space.

*Table 9.* **Ablation of different kernels in GP experts.** Results are reported on the MBO task.

| Method | SQ (%) | AUOC | APD |
|---|---|---|---|
| RBF Kernel | 149.06 ± 13.08 | 1.183 ± 0.194 | 0.245 ± 0.021 |
| Additive Kernel | 161.46 ± 61.78 | 1.142 ± 0.320 | 0.213 ± 0.060 |

**Ablation of PIEVO Dynamics and Test-time Reasoning.** We further analyze the system's internal mechanisms (see Appendix G and test-time reasoning by `Think` mode in F). We found that: (a) PIEVO systematically evolves high-fidelity principles that correlate directly with solution quality, validating the efficacy of principle-space optimization. (b) a *cognitive interference* effect where `Think` models degrade performance ($\downarrow 26.35\%$), as LLM's diverse reasoning conflicts with the rigorous utility of strategic layers.

## 5. Conclusion

We presented PIEVO, a framework that reformulates scientific discovery from static hypothesis search to Bayesian optimization over an evolving principle space. By synergizing *Information-Directed Selection* via GP experts with *Anomaly-Driven Epistemic Augmentation*, PIEVO actively refines its theoretical worldview. Empirically, it achieves a 31.06% gain in solution quality, sublinear regret, and autonomous discovery of chiral physics, establishing principle evolution as a key paradigm for scientific agents.

## Acknowledgement

This work was supported in part by the National Natural Science Foundation of China (NSFC) under No. 62576285, No. 92356310, No. 22575200, Research Center for Industries of the Future (RCIF) at Westlake University, and Westlake Education Foundation.

We thank the anonymous reviewers for their insightful comments and suggestions, which significantly improved this paper. We also express our gratitude to Dr. Tengjie Lyu and Dr. Jie Wu for their valuable discussions and feedback.

## Impact Statement

This paper presents work whose goal is to advance the field of automated scientific discovery. There are many potential societal consequences of our work, none which we feel must be specifically highlighted here.

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

## A. Theory with Coherent Principle Augmentation

We formalize a learning process that (i) selects hypotheses guided by candidate principles and (ii) *coherently* augments the principle space when anomalies appear, while maintaining a single fixed prior over a (possibly countable) *universal* principle space. This section establishes regret guarantees and posterior consistency under explicit, verifiable assumptions of PⅠEᴠᴏ. The analysis separates the contributions of identification (principle selection) and principle-to-hypothesis execution, and makes the cost of discovery delay, i.e., the time until an adequate principle becomes available.

### A.1. Assumptions

**Universal principle space and coherent augmentation.** Let $\bar{\mathcal{P}}$ be a countable *universe of principles* endowed with a fixed prior $p_0$ such that

$$H(p_0) := H(P) = -\sum_{P \in \bar{\mathcal{P}}} p_0(P) \ln p_0(P) < \infty,$$

where $P \sim p_0$ is the (a priori) random principle and $H(\cdot)$ denotes Shannon entropy. At each round $t = 1, 2, \ldots$, PⅠEᴠᴏ maintains a *working set* $\mathcal{P}_t \subseteq \bar{\mathcal{P}}$ of principles. It may *augment* $\mathcal{P}_t$ by adding new elements of $\bar{\mathcal{P}}$ in response to anomalies. Augmentation is *Bayesian coherent* in the following sense.

**Assumption 1** (Coherent augmentation). *For every $t$, the posterior over all $P \in \bar{\mathcal{P}}$ is defined by Bayes' rule from the fixed prior $p_0$ and the full history $H_t$:*

$$p_t(P) := p(P \mid H_t) \propto p_0(P) \prod_{s=1}^{t} p(y_s \mid h_s, P).$$

*When new principles are added to the working set at time $t$, their (stored) posterior equals the restriction of $p_t(\cdot)$ to those elements. No local re-initialization of priors occurs.*

**Observation model, rewards, and regret.** On round $t$, PⅠEᴠᴏ selects a principle $P_t \in \mathcal{P}_t$, then a hypothesis $h_t \in \mathcal{H}$ (via a generator $\pi_t$), and receives $y_t \sim p(\cdot \mid h_t, P^\star)$ for a fixed but unknown $P^\star \in \bar{\mathcal{P}}$. A bounded reward $r : \mathcal{H} \times \bar{\mathcal{P}} \to [0, 1]$ is realized, and regret is measured against the *oracle hypothesis under the true principle*:

$$v^\star = \max_{h \in \mathcal{H}} r(h, P^\star), \qquad \Delta_t = v^\star - \mathbb{E}\left[r(h_t, P^\star) \mid H_{t-1}\right], \qquad \mathrm{Regret}(T) = \sum_{t=1}^{T} \Delta_t.$$

Define $h^\star(P) \in \arg\max_h r(h, P)$ and the uniform deviation bound:

$$G := \sup_{h \in \mathcal{H}, \ P \in \bar{\mathcal{P}}} \left| r\left(h^\star(P), P^\star\right) - r(h, P^\star) \right| \leq 1.$$

**Identifiability and persistent information.**

**Assumption 2** (Identifiability). *For every $P \neq P^\star$ there exists $h \in \mathcal{H}$ such that $p(\cdot \mid h, P) \neq p(\cdot \mid h, P^\star)$ (e.g., in total variation or KL).*

**Assumption 3** (Persistent information). *Conditioned on any history, the policy selects actions that (with positive probability, infinitely often) yield strictly positive information about each wrong $P \neq P^\star$; equivalently, the cumulative expected information against any $P \neq P^\star$ diverges almost surely once an informative discriminator is available in the working set.*

**Principle-to-hypothesis generator.** Given a chosen principle $P_t$, a generator (e.g., a well-trained large language model in our case) produces a finite candidate set $\mathcal{C}_t(P_t) \subset \mathcal{H}$ with cardinality $N_t := |\mathcal{C}_t(P_t)| \geq 2$ and a proposal distribution $\pi_t(\cdot \mid P_t, H_{t-1})$ supported on $\mathcal{C}_t(P_t)$. Let $H(\pi_t)$ denote its entropy. We assume:

(G1) *Low-entropy proposal:* almost surely $H(\pi_t) \leq \varepsilon_t$, with $0 \leq \varepsilon_t \leq \ln N_t$.

(G2) *Candidate-pool approximation:* almost surely there exists $h_t^\dagger \in \mathcal{C}_t(P_t)$ such that

$$r\left(h_t^\dagger, P^\star\right) \geq r\left(h^\star(P_t), P^\star\right) - \delta_t, \qquad \delta_t \geq 0.$$

(G3) *Calibration of top mass:* almost surely $\pi_t(h_t^\dagger) \geq \alpha_t$, where $\alpha_t$ is the top mass of $\pi_t$ on $\mathcal{C}_t(P_t)$.

**Information and IDS selection.** Let $U_t^{\mathrm{EP}} := H(P \mid H_{t-1})$ be the (Bayesian) posterior entropy of the principle before seeing $y_t$, and

$$I_t \ := \ \mathbb{E}_{Y_t \mid H_{t-1}, h_t} \left[ U_t^{\mathrm{EP}} - U_{t+1}^{\mathrm{EP}} \right] \ = \ I\left(P; Y_t \mid H_{t-1}, h_t\right)$$

be the expected entropy reduction (mutual information). The action $h_t$ is chosen to satisfy an information-directed sampling (IDS) ratio bound.

**Assumption 4** (IDS ratio). *For every t, either $I_t = 0$ and $\Delta_t^{(\mathrm{ID})} = 0$, or $I_t > 0$ and*

$$\frac{\left(\Delta_t^{(\mathrm{ID})}\right)^2}{I_t} \ \leq \ \Gamma \qquad \textit{for some fixed } \Gamma > 0,$$

*where the ID-regret component is*

$$\Delta_t^{(\mathrm{ID})} \ := \ \mathbb{E}\left[r\left(h^\star(P^\star), P^\star\right) - r\left(h^\star(P_t), P^\star\right) \mid H_{t-1}\right].$$

**Discovery time and discoverability.** Define the *discovery time* of an adequate principle as

$$\tau_{\mathrm{disc}} \ := \ \inf\left\{t \geq 1: \ \mathcal{P}_t \text{ contains } P^\star \text{ or some } P \text{ with } \sup_{h \in \mathcal{H}} \left\|p(\cdot \mid h, P) - p(\cdot \mid h, P^\star)\right\|_{\mathrm{TV}} \ \leq \ \varepsilon\right\},$$

for a fixed accuracy parameter $\varepsilon \in [0, 1)$. Before $\tau_{\mathrm{disc}}$, the working set may not allow informative discrimination of $P^\star$.

**Assumption 5** (Discoverability). *There exist $S \in \mathbb{N}$ and $\rho \in (0, 1]$ such that the augmentation routine is triggered at least once every $S$ rounds and, whenever triggered, it proposes a new principle in the above $\varepsilon$-ball with probability at least $\rho$ (possibly via multiple draws per trigger). Then $\mathbb{E}[\tau_{\mathrm{disc}}] \leq S/\rho$.*

### A.2. Auxiliary Lemmas

**Lemma A.1** (Regret decomposition). *For every t,*

$$\Delta_t \ = \ \underbrace{\mathbb{E}\left[r\left(h^\star(P^\star), P^\star\right) - r\left(h^\star(P_t), P^\star\right) \mid H_{t-1}\right]}_{\Delta_t^{(\mathrm{ID})}} + \underbrace{\mathbb{E}\left[r\left(h^\star(P_t), P^\star\right) - r(h_t, P^\star) \mid H_{t-1}\right]}_{\Delta_t^{(\mathrm{PH})}}.$$

*Proof.* Condition on $H_{t-1}$. Decompose the instantaneous gap between $v^\star$ and the expected reward of the chosen hypothesis $h_t$ by introducing the intermediate comparator $h^\star(P_t)$:

$$\begin{aligned} \Delta_t &= v^\star - \mathbb{E}[r(h_t, P^\star) \mid H_{t-1}] \\ &= \left(v^\star - \underbrace{\mathbb{E}[r(h^\star(P_t), P^\star) \mid H_{t-1}]}_{Intermediate}\right) + \left(\underbrace{\mathbb{E}[r(h^\star(P_t), P^\star) \mid H_{t-1}]}_{Intermediate} - \mathbb{E}[r(h_t, P^\star) \mid H_{t-1}]\right) \\ &= \Delta_t^{(\mathrm{ID})} + \Delta_t^{(\mathrm{PH})}, \end{aligned}$$

which is the stated decomposition. $\qquad\square$

**Lemma A.2** (Discrete-entropy bound for PH-regret). *Under (G1)–(G3), conditioning on $H_{t-1}$ and $P_t$, let $N_t = |\mathcal{C}_t(P_t)| \geq 2$, let $\varepsilon_t \in [0, \ln N_t]$ be such that $H(\pi_t) \leq \varepsilon_t$, and let*

$$\Phi_N(\alpha) \ := \ H_{\mathrm{b}}(\alpha) + (1 - \alpha)\ln(N - 1), \qquad \alpha_N(\varepsilon) \ := \ \max\{\alpha \in [1/N, 1]: \ \Phi_N(\alpha) = \varepsilon\},$$

$$\varphi_N(\varepsilon) \ := \ 1 - \alpha_N(\varepsilon).$$

*Then*

$$\Delta_t^{(\mathrm{PH})} \ \leq \ \delta_t \ + \ G \cdot \varphi_{N_t}(\varepsilon_t).$$

*Proof.* Condition on $H_{t-1}$ and $P_t$. By definition,

$$\Delta_t^{(\mathrm{PH})} = \sum_{h \in \mathcal{C}_t} \pi_t(h)\left(r(h^\star(P_t), P^\star) - r(h, P^\star)\right).$$

Split the sum at $h_t^\dagger$:

$$\Delta_t^{\text{(PH)}} = \pi_t(h_t^\dagger)\Big(r(h^\star(P_t), P^\star) - r(h_t^\dagger, P^\star)\Big) + \sum_{h \neq h_t^\dagger} \pi_t(h)\Big(r(h^\star(P_t), P^\star) - r(h, P^\star)\Big)$$

$$\leq \pi_t(h_t^\dagger) \cdot \delta_t + (1 - \pi_t(h_t^\dagger)) \cdot G \leq \delta_t + G \cdot (1 - \pi_t(h_t^\dagger)).$$

Let $\alpha_t := \max_h \pi_t(h)$ be the top mass. By calibration, $\pi_t(h_t^\dagger) \geq \alpha_t$. Among distributions on $N_t$ atoms with top mass $\alpha$, entropy is maximized by spreading $1 - \alpha$ uniformly on the remaining $N_t - 1$ atoms, giving $\Phi_{N_t}(\alpha)$. Hence

$$H(\pi_t) \leq \Phi_{N_t}(\alpha_t) \leq \varepsilon_t \quad \Rightarrow \quad \alpha_t \geq \alpha_{N_t}(\varepsilon_t), \quad 1 - \pi_t(h_t^\dagger) \leq 1 - \alpha_t \leq \varphi_{N_t}(\varepsilon_t).$$

Combine with the previous inequality to conclude. $\qquad\square$

**Lemma A.3** (IDS information-ratio bound). *If Assumption 4 holds, then for any $T \geq 1$,*

$$\sum_{t=1}^T \Delta_t^{\text{(ID)}} \leq \sqrt{\Gamma T \sum_{t=1}^T I_t}.$$

*Taking expectations preserves the bound, and using $\sum_{t=1}^T I_t \leq H(P)$ yields*

$$\mathbb{E}\left[\sum_{t=1}^T \Delta_t^{\text{(ID)}}\right] \leq \sqrt{\Gamma T H(P)}.$$

*Proof.* For indices with $I_t = 0$ we also have $\Delta_t^{\text{(ID)}} = 0$ by assumption. For the others, $(\Delta_t^{\text{(ID)}})^2 \leq \Gamma I_t$. Then

$$\Big(\sum_{t=1}^T \Delta_t^{\text{(ID)}}\Big)^2 = \Big(\sum_{t=1}^T \sqrt{\tfrac{(\Delta_t^{\text{(ID)}})^2}{I_t}} \sqrt{I_t}\Big)^2 \leq \Big(\sum_{t=1}^T \tfrac{(\Delta_t^{\text{(ID)}})^2}{I_t}\Big)\Big(\sum_{t=1}^T I_t\Big) \leq \Gamma T \sum_{t=1}^T I_t,$$

by Cauchy–Schwarz and the per-round ratio bound. Take square roots. For expectations, use Jensen's inequality for $\sqrt{\cdot}$ and the information budget bound in Lemma A.4 below. $\qquad\square$

**Lemma A.4** (Information accounting under coherent augmentation). *Under Assumption 1, for every $T \geq 1$,*

$$\sum_{t=1}^T I_t = H(P) - H(P \mid H_T) \qquad \text{almost surely,}$$

*and consequently $\mathbb{E}\left[\sum_{t=1}^T I_t\right] = H(P) - \mathbb{E}\left[H(P \mid H_T)\right] \leq H(P)$.*

*Proof.* Fix an outcome path. Coherence ensures that at *every* round the posterior $p_t(\cdot)$ equals the Bayes posterior on $\bar{\mathcal{P}}$ based on $H_t$. By the chain rule for entropy and mutual information,

$$H(P \mid H_{t-1}) - \mathbb{E}\left[H(P \mid H_t) \mid H_{t-1}\right] = I\left(P; Y_t \mid H_{t-1}, h_t\right) = I_t.$$

Summing from $t = 1$ to $T$ telescopes:

$$\sum_{t=1}^T I_t = \sum_{t=1}^T \big(H(P \mid H_{t-1}) - H(P \mid H_t)\big) = H(P \mid H_0) - H(P \mid H_T) = H(P) - H(P \mid H_T).$$

Taking expectations yields the second statement; the inequality follows since entropy is nonnegative. $\qquad\square$

**Lemma A.5** (Discovery delay cost). *For any $T \geq 1$,*

$$\sum_{t=1}^{T} \Delta_t \leq G \cdot \min\{T, \tau_{\text{disc}} - 1\} + \sum_{t=\tau_{\text{disc}}}^{T} \left( \Delta_t^{(\text{PH})} + \Delta_t^{(\text{ID})} \right).$$

*Consequently,* $\mathbb{E}\left[\sum_{t=1}^{\min\{T,\tau_{\text{disc}}-1\}} \Delta_t \right] \leq G\mathbb{E}\left[\min\{T, \tau_{\text{disc}} - 1\}\right] \leq G\mathbb{E}[\tau_{\text{disc}}]$. *If Assumption 5 holds, then* $\mathbb{E}[\tau_{\text{disc}}] \leq S/\rho$.

*Proof.* Before $\tau_{\text{disc}}$ the algorithm cannot (by definition) rely on a principle in $\mathcal{P}_t$ that is $\varepsilon$-adequate for $P^\star$. The per-round regret is bounded by $G$. Sum the bound up to $\min\{T, \tau_{\text{disc}} - 1\}$ and apply Lemma A.1 for the remainder. The expectation bound follows from monotonicity of expectation. The discoverability bound is a standard geometric waiting-time estimate under Assumption 5. $\square$

## A.3. Regret Bound and Posterior Concentration

Define the per-round PH control

$$\psi_{N_t}(\varepsilon_t, \delta_t) := \delta_t + G\varphi_{N_t}(\varepsilon_t),$$

where $\varphi_{N_t}$ is from Lemma A.2.

**Theorem A.6** (Regret with coherent augmentation). *Suppose bounded rewards, Assumptions 1, 2, 4, and generator conditions (G1)–(G3) hold. Then for every horizon $T \geq 1$,*

$$\mathbb{E}\left[\text{Regret}(T)\right] \leq \sum_{t=1}^{T} \mathbb{E}\left[\psi_{N_t}(\varepsilon_t, \delta_t)\right] + \sqrt{\Gamma T H(p_0)} + G\mathbb{E}[\tau_{\text{disc}}].$$

*If in addition $\sum_{t=1}^{\infty} \mathbb{E}[\psi_{N_t}(\varepsilon_t, \delta_t)] < \infty$ and Assumption 5 holds (so $\mathbb{E}[\tau_{\text{disc}}] < \infty$), then $\mathbb{E}[\text{Regret}(T)] = \mathcal{O}(\sqrt{T})$.*

*Proof.* Fix $T$. Apply Lemma A.5 to split the sum at $\tau_{\text{disc}}$:

$$\text{Regret}(T) \leq G \cdot \min\{T, \tau_{\text{disc}} - 1\} + \sum_{t=\tau_{\text{disc}}}^{T} \Delta_t^{(\text{PH})} + \sum_{t=\tau_{\text{disc}}}^{T} \Delta_t^{(\text{ID})}.$$

Take expectations and use $\mathbb{E}[\min\{T, \tau_{\text{disc}} - 1\}] \leq \mathbb{E}[\tau_{\text{disc}}]$ for the first term. For the PH part, Lemma A.2 gives $\Delta_t^{(\text{PH})} \leq \psi_{N_t}(\varepsilon_t, \delta_t)$ pointwise, hence

$$\mathbb{E}\left[ \sum_{t=\tau_{\text{disc}}}^{T} \Delta_t^{(\text{PH})} \right] \leq \sum_{t=1}^{T} \mathbb{E}\left[\psi_{N_t}(\varepsilon_t, \delta_t)\right].$$

For the ID part, Lemma A.3 yields

$$\sum_{t=\tau_{\text{disc}}}^{T} \Delta_t^{(\text{ID})} \leq \sqrt{\Gamma(T - \tau_{\text{disc}} + 1) \sum_{t=\tau_{\text{disc}}}^{T} I_t} \leq \sqrt{\Gamma T \sum_{t=1}^{T} I_t},$$

where the second inequality uses $T - \tau_{\text{disc}} + 1 \leq T$ and monotonicity of the sum of informations. Taking expectations and invoking Lemma A.4 gives

$$\mathbb{E}\left[ \sum_{t=\tau_{\text{disc}}}^{T} \Delta_t^{(\text{ID})} \right] \leq \sqrt{\Gamma T \mathbb{E}\left[ \sum_{t=1}^{T} I_t \right]} \leq \sqrt{\Gamma T H(p_0)}.$$

Combine the three bounds to conclude. $\square$

**Corollary A.7** (Sublinear regret). *Under the assumptions of Theorem A.6, if $P^\star \in \mathcal{P}_0$ (so $\tau_{\text{disc}} = 1$) or more generally $\mathbb{E}[\tau_{\text{disc}}] < \infty$, and $\sum_{t=1}^{\infty} \mathbb{E}[\psi_{N_t}(\varepsilon_t, \delta_t)] < \infty$, then $\mathbb{E}[\text{Regret}(T)] = \mathcal{O}(\sqrt{T})$.*

**Theorem A.8** (Posterior concentration). *Under Assumptions 1, 2, and 3, $\mathbb{E}[H(P \mid H_T)] \to 0$ as $T \to \infty$. If $|\bar{\mathcal{P}}| < \infty$, then for any measurable estimator $\hat{P}_T = \hat{P}(H_T)$, $\Pr(\hat{P}_T \neq P^\star) \to 0$.*

*Proof.* Lemma A.4 gives $\mathbb{E}\left[\sum_{t=1}^{T} I_t\right] = H(p_0) - \mathbb{E}[H(P \mid H_T)]$. Under Assumption 3, the cumulative expected information diverges to $H(p_0)$, thus $\mathbb{E}[H(P \mid H_T)] \to 0$. For finite $\bar{\mathcal{P}}$, Fano's inequality implies $\mathbb{E}[H(P \mid H_T)] \leq H_{\mathrm{b}}(\Pr(\hat{P}_T \neq P^\star)) + \Pr(\hat{P}_T \neq P^\star)\ln(|\bar{\mathcal{P}}| - 1)$, which forces the error probability to vanish. $\qquad\square$

> **Takeaway (Regret Bound of PIEVO).** The derivation of Theorem A.6 reveals a fundamental efficiency property. Our analysis rests on a three-step logic: (i) decomposing regret into identification (ID), execution (PH), and discovery delay components; (ii) leveraging *coherent augmentation* to telescope the information gain $\sum I_t$ against a fixed prior entropy $H(p_0)$, regardless of when principles are added; and (iii) bounding the generator's deviation via entropy constraints. The resulting $\tilde{\mathcal{O}}(\sqrt{T})$ bound confirms that the primary bottleneck is not the size of the search space, but the *discovery time* of the true principle. In the following section, we refine this analysis to quantify the specific sample complexity required to drive the identification error to zero.

## A.4. Precise Identification and Sample-Complexity Statements

In Theorem A.8, we found that posterior mass on wrong principles converges to 0, The next proposition quantifies an exponential decay (precisely) of posterior mass on any fixed wrong principle once informative actions are taken with nontrivial frequency.

**Proposition A.9** (Exponential decay of wrong principles). *Fix $P \neq P^\star$. Define the conditional expected KL at round $t$,*

$$\kappa_t(P) := \mathbb{E}\left[D_{\mathrm{KL}}\left(p(\cdot \mid h_t, P^\star)\|p(\cdot \mid h_t, P)\right)|H_{t-1}\right].$$

*If there exist $\underline{\kappa} > 0$ and $\eta \in (0, 1]$ such that with probability at least $\eta$ (infinitely often) we have $\kappa_t(P) \geq \underline{\kappa}$, then*

$$\mathbb{E}\left[\log \frac{p_t(P)}{p_t(P^\star)}\right] \leq \log \frac{p_0(P)}{p_0(P^\star)} - \underline{\kappa}\mathbb{E}[N_t],$$

*where $N_t$ is the number of rounds up to $t$ at which the event $\{\kappa_s(P) \geq \underline{\kappa}\}$ occurs. In particular, $\mathbb{E}[p_t(P)]$ decays to 0.*

*Proof.* Let $p_t(P) \triangleq p(P \mid H_{t-1})$ be the posterior at the start of round $t$. The posterior after $t$ rounds, $p_{t+1}(P) \triangleq p(P \mid H_t)$, is given by Bayes' rule:

$$p_{t+1}(P) \propto p_t(P) \cdot p(y_t \mid h_t, P)$$

The log-likelihood ratio relative to the true principle $P^\star$ evolves as:

$$\log \frac{p_{t+1}(P)}{p_{t+1}(P^\star)} = \log \frac{p_t(P)}{p_t(P^\star)} + \log \frac{p(y_t \mid h_t, P)}{p(y_t \mid h_t, P^\star)}.$$

Unrolling this recursion back to the prior $p_0$ gives:

$$\log \frac{p_{t+1}(P)}{p_{t+1}(P^\star)} = \log \frac{p_0(P)}{p_0(P^\star)} + \sum_{s=1}^{t} \log \frac{p(y_s \mid h_s, P)}{p(y_s \mid h_s, P^\star)}.$$

We take the total expectation, $\mathbb{E}[\cdot]$, which is over the joint distribution of the full history $H_t = \{(h_s, y_s)\}_{s=1}^{t}$ generated under $P^\star$ and the algorithm's (possibly random) choices. We use the tower property of expectation $\mathbb{E}[\cdot] = \mathbb{E}_{H_{s-1}}[\mathbb{E}_{Y_s, h_s}[\cdot \mid H_{s-1}]]$. Let $Z_s = \log \frac{p(y_s|h_s, P)}{p(y_s|h_s, P^\star)}$. Its conditional expectation given the past $H_{s-1}$ is:

$$\mathbb{E}[Z_s \mid H_{s-1}] = \mathbb{E}_{h_s \sim \pi_s(\cdot|H_{s-1})}\left[\mathbb{E}_{Y_s \sim p(\cdot|h_s, P^\star)}\left[\log \frac{p(Y_s \mid h_s, P)}{p(Y_s \mid h_s, P^\star)} \mid h_s, H_{s-1}\right] \mid H_{s-1}\right]$$

$$= \mathbb{E}_{h_s \sim \pi_s(\cdot|H_{s-1})}\left[-D_{\mathrm{KL}}\left(p(\cdot \mid h_s, P^\star)\|p(\cdot \mid h_s, P)\right) \mid H_{s-1}\right]$$

$$= -\kappa_s(P).$$

Taking the total expectation of the unrolled sum:

$$\mathbb{E}\left[\log \frac{p_{t+1}(P)}{p_{t+1}(P^\star)}\right] = \log \frac{p_0(P)}{p_0(P^\star)} + \sum_{s=1}^{t} \mathbb{E}\left[\mathbb{E}[Z_s \mid H_{s-1}]\right]$$

$$= \log \frac{p_0(P)}{p_0(P^\star)} - \sum_{s=1}^{t} \mathbb{E}[\kappa_s(P)].$$

Notably, the proposition statement uses $p_t(P)$, which we interpret as the posterior after $t$ rounds, i.e., $p_{t+1}(P)$ in our derivation. We proceed with $t$ as the final time index. Let $N_t = \sum_{s=1}^{t} \mathbf{1}_{\{\kappa_s(P) \geq \underline{\kappa}\}}$ be the number of rounds where the KL is sufficiently large. We can lower-bound the sum of expected KL terms:

$$\sum_{s=1}^{t} \mathbb{E}[\kappa_s(P)] = \mathbb{E}\left[\sum_{s=1}^{t} \kappa_s(P)\right] \geq \mathbb{E}\left[\sum_{s=1}^{t} \kappa_s(P) \cdot \mathbf{1}_{\{\kappa_s(P) \geq \underline{\kappa}\}}\right] \geq \mathbb{E}\left[\sum_{s=1}^{t} \underline{\kappa} \cdot \mathbf{1}_{\{\kappa_s(P) \geq \underline{\kappa}\}}\right] = \underline{\kappa}\mathbb{E}[N_t].$$

Substituting this back gives the first claim:

$$\mathbb{E}\left[\log \frac{p_t(P)}{p_t(P^\star)}\right] \leq \log \frac{p_0(P)}{p_0(P^\star)} - \underline{\kappa}\mathbb{E}[N_t].$$

For the second claim, let $R_t(P) = p_t(P)/p_t(P^\star)$. The above bound shows that if $\mathbb{E}[N_t] \to \infty$ (which is implied by the *infinitely often* assumption), then $\mathbb{E}[\log R_t(P)] \to -\infty$. This implies $R_t(P) \to 0$ almost surely (e.g., by the supermartingale convergence theorem). Since $p_t(P) = \frac{R_t(P)}{1 + R_t(P) + \sum_{P' \neq P, P^\star} R_t(P')} \leq R_t(P)$, we have $p_t(P) \to 0$ almost surely. Because $p_t(P)$ is bounded in $[0, 1]$, the Bounded Convergence Theorem applies, and we conclude $\mathbb{E}[p_t(P)] \to \mathbb{E}[0] = 0$. $\qquad\square$

**Proposition A.10** ($\varepsilon$-optimal average regret sample complexity). *Under the conditions of Theorem A.6, let $\bar{R}_T := \mathbb{E}[\text{Regret}(T)]/T$. Then for all $T \geq 1$,*

$$\bar{R}_T \leq \frac{1}{T}\sum_{t=1}^{T} \mathbb{E}[\psi_{N_t}(\varepsilon_t, \delta_t)] + \sqrt{\frac{\Gamma H(p_0)}{T}} + \frac{G\mathbb{E}[\tau_{\text{disc}}]}{T}.$$

*Hence, for any $\varepsilon > 0$, it suffices to take*

$$T \gtrsim \frac{\Gamma H(p_0)}{\varepsilon^2} + \frac{G\mathbb{E}[\tau_{\text{disc}}]}{\varepsilon}$$

*to ensure $\bar{R}_T \leq \varepsilon$ up to the (summable) PH term.*

*Proof.* The first inequality is obtained by dividing the bound in Theorem A.6 by $T$.

$$\bar{R}_T = \frac{\mathbb{E}[\text{Regret}(T)]}{T} \leq \frac{1}{T}\sum_{t=1}^{T} \mathbb{E}[\psi_{N_t}(\varepsilon_t, \delta_t)] + \frac{\sqrt{\Gamma T H(p_0)}}{T} + \frac{G\mathbb{E}[\tau_{\text{disc}}]}{T}.$$

Simplifying $\sqrt{\Gamma T H(p_0)}/T = \sqrt{\Gamma H(p_0)/T}$ gives the stated bound.

For the second part, we want to find $T$ such that $\bar{R}_T \leq \varepsilon$. The PH term $\frac{1}{T}\sum_{t=1}^{T} \mathbb{E}[\psi_{N_t}(\varepsilon_t, \delta_t)]$ vanishes as $T \to \infty$ because the sum is assumed to be finite (summable). We therefore seek $T$ such that the remaining two terms are bounded by $\varepsilon$:

$$\sqrt{\frac{\Gamma H(p_0)}{T}} + \frac{G\mathbb{E}[\tau_{\text{disc}}]}{T} \leq \varepsilon.$$

A sufficient condition is to make each term individually less than or equal to $\varepsilon/2$:

1. For the identification term:

$$\sqrt{\frac{\Gamma H(p_0)}{T}} \leq \frac{\varepsilon}{2} \implies \frac{\Gamma H(p_0)}{T} \leq \frac{\varepsilon^2}{4} \implies T \geq \frac{4\Gamma H(p_0)}{\varepsilon^2}.$$

2. For the discovery delay term:

$$\frac{G\mathbb{E}[\tau_{\text{disc}}]}{T} \leq \frac{\varepsilon}{2} \implies T \geq \frac{2G\mathbb{E}[\tau_{\text{disc}}]}{\varepsilon}.$$

To satisfy both conditions, $T$ must be larger than the maximum of these two lower bounds. This scaling is captured by the $\gtrsim$ (or $\mathcal{O}$) notation, which hides the constants:

$$T \gtrsim \frac{\Gamma H(p_0)}{\varepsilon^2} + \frac{G\mathbb{E}[\tau_{\text{disc}}]}{\varepsilon}.$$

This shows the required sample complexity to achieve an $\varepsilon$-optimal average regret. $\qquad\square$

**Discussion of the PH term.** The quantity $\sum_t \mathbb{E}[\psi_{N_t}(\varepsilon_t, \delta_t)]$ is algorithm-dependent and interprets as the cumulative price of keeping the generator *low-entropy* yet *well-calibrated* and of ensuring candidate pools contain near-optimal actions (relative to the currently selected principle). Constraining decoding (e.g., top-$k$), temperature control, and explicit pruning may be used to guarantee $H(\pi_t) \leq \varepsilon_t$ while maintaining the calibration property; these design choices directly control the PH component without impacting the information budget $H(p_0)$ thanks to coherence.

---

**Takeaway (Propositions of Sample Complexity Required to Drive the Identification Error to Zero).** Coherent augmentation preserves the global information accounting $\sum_{t=1}^T I_t = H(p_0) - H(P \mid H_T)$, which in turn keeps the IDS-driven identification regret at $\tilde{O}(\sqrt{T})$ with budget $H(p_0)$. Exploration at the principle level only adds an explicit, interpretable *discovery-delay* term $G\mathbb{E}[\tau_{\text{disc}}]$, while the PH term remains controlled by generator entropy and calibration. Under identifiability and persistent information, the posterior concentrates on the true principle and misidentification probability vanishes (finite universes), completing a tight exploration–exploitation loop over an expanding but coherently managed principle space.

---

**Remark on the Gap between Theory and Algorithm.** The theoretical model above assumes a hierarchical selection process where a principle $P_t$ is sampled first, followed by a hypothesis $h_t$. In contrast, the practical implementation (Algorithm 1) employs a marginalized Information-Directed Sampling (IDS) strategy, where $\mathcal{A}_H$ selects $h_t$ by marginalizing over the posterior $p_t(P)$. We emphasize that this algorithmic choice is consistent with the theoretical objective: by selecting $h$ to minimize the ratio $\Delta_t^2 / I_t(P; Y_t)$, the algorithm is directly minimizing the cumulative identification regret $\sum \Delta_t^{(\text{ID})}$. The marginalization serves as a robust estimator for the expected information gain $I_t$ in the presence of principle uncertainty, effectively maintaining the $\mathcal{O}(\sqrt{T})$ identification rate while smoothing the exploration in the principle space.

## B. Theoretical Justification of Naive Feature Design in Gaussian Process Experts

We provide the formal proof that the structural feature embedding $\phi(h, P)$, defined in Equation 2, constitutes a valid and sufficient input space for the Gaussian Process experts to learn the likelihood $p(y \mid h, P)$.

**Proposition B.1** (Geometric Sufficiency and Kernel Validity). *Let $\mathcal{E}$ be a Hilbert space where principles and hypotheses are represented as **normalized** semantic vectors $\boldsymbol{e}_P, \boldsymbol{e}_h \in \mathcal{E}$ (i.e., $\|\boldsymbol{e}_P\| = \|\boldsymbol{e}_h\| = 1$). If the latent reward function $f(h, P)$ depends continuously on the semantic alignment between $h$ and $P$, then the mapping $\phi(h, P) = [\langle \boldsymbol{e}_h, \boldsymbol{e}_P \rangle, \|\boldsymbol{e}_h - \boldsymbol{e}_P\|_2]^\top$ forms a sufficient statistic for this alignment, and the induced kernel $k_\phi\big((h, P), (h', P')\big)$ is a valid positive semi-definite covariance function.*

*Proof.* **1. Geometric decomposition of semantic alignment.** We posit that the conditional probability $p(y \mid h, P)$ is governed by the semantic compatibility between the hypothesis $h$ and the principle $P$. In the embedding space $\mathcal{E}$, the relationship between any two vectors is fully characterized by their magnitudes and the angle $\theta$ between them. Given the normalization assumption ($\|\boldsymbol{e}_h\| = \|\boldsymbol{e}_P\| = 1$), the geometry of the pair reduces to a single degree of freedom: the angle $\theta$. By the law of cosines, the Euclidean distance is determined solely by this angle:

$$\|\boldsymbol{e}_h - \boldsymbol{e}_P\|_2^2 = 2 - 2\langle \boldsymbol{e}_h, \boldsymbol{e}_P \rangle \tag{5}$$

Consequently, the pair of scalars $(\langle \boldsymbol{e}_h, \boldsymbol{e}_P \rangle, \|\boldsymbol{e}_h - \boldsymbol{e}_P\|_2)$ contains redundant but complete information to reconstruct the geometric configuration of $(\boldsymbol{e}_h, \boldsymbol{e}_P)$ up to rigid rotation. Therefore, any function $f$ dependent on the relative geometry of $h$ and $P$ can be factorized as $f(h, P) = g(\phi(h, P))$ for some function $g : \mathbb{R}^2 \to \mathbb{R}$.

**2. Kernel validity.** A Gaussian Process requires a positive semi-definite (PSD) kernel function. Let $\kappa : \mathbb{R}^d \times \mathbb{R}^d \to \mathbb{R}$ be a standard PSD kernel (e.g., RBF). We define the kernel on the joint hypothesis-principle space $\mathcal{X} = \mathcal{H} \times \bar{\mathcal{P}}$ as:

$$k\big((h, P), (h', P')\big) \triangleq \kappa\big(\phi(h, P), \phi(h', P')\big) \tag{6}$$

Since $\phi : \mathcal{X} \to \mathbb{R}^2$ is a deterministic feature map, the resulting kernel $k$ is a valid kernel on $\mathcal{X}$ (specifically, it is the pullback of $\kappa$ along $\phi$). This ensures the GP expert $\mathcal{M}_P$ defines a valid prior distribution over functions on the complex semantic space $\mathcal{H} \times \bar{\mathcal{P}}$. $\square$

## C. The Comparison of Regret Bound: PIEVO and `PiFlow`

In this section, we analyze the theoretical efficiency of PIEVO compared to the existing principle-guided framework `PiFlow` (Pu et al., 2025). We demonstrate that by decomposing the search space into an abstract principle layer and a concrete hypothesis layer, **PIEVO achieves significantly faster convergence** through a reduction in the complexity of the identification task.

**Regret decomposition and convergence in PIEVO.** The design of the loop directly maps to our theoretical regret decomposition. The instantaneous regret $\Delta_t = v^\star - \mathbb{E}[r(h_t, P^\star) \mid H_{t-1}]$ can be partitioned as:

$$\Delta_t = \underbrace{\left( v^\star - \mathbb{E}[r(h^\star(P_t), P^\star) \mid H_{t-1}] \right)}_{\Delta_t^{(\mathrm{ID})}:\text{Identifying principle}} + \underbrace{\left( \mathbb{E}[r(h^\star(P_t), P^\star) \mid H_{t-1}] - \mathbb{E}[r(h_t, P^\star) \mid H_{t-1}] \right)}_{\Delta_t^{(\mathrm{PH})}:\text{Principle-to-hypothesis}}$$

where $\Delta_t^{(\mathrm{ID})}$ captures the loss from not yet identifying $P^\star$, and $\Delta_t^{(\mathrm{PH})}$ captures the loss from imperfect hypothesis generation. The cumulative regret is bounded as:

$$\mathbb{E}[\mathrm{Regret}(T)] \leq \underbrace{\sum_{t=1}^{T} \mathbb{E}[\psi_{N_t}(\varepsilon_t, \delta_t)]}_{\text{Hypothesis Generation Cost}} + \underbrace{\sqrt{\Gamma T H(P)}}_{\text{Principle Identification Cost}}$$

If the hypothesis generation cost is $o(T)$, the dominant term scales as $\sqrt{\Gamma H(P)}\sqrt{T}$, yielding $O(\sqrt{T})$ convergence. Crucially, the identification cost depends on the entropy $H(P)$ of the principle space.

**Regret in PiFlow.** As illustrates in PiFlow (Pu et al., 2025), the regret is bounded by $\mathbb{E}[\mathrm{Regret}(T)] \leq \sum_{t=1}^{T} \mathbb{E}[\psi_{N_t}(\varepsilon_t, \delta_t)] + \sqrt{\Gamma T \log |\mathcal{H}|}$, where $\mathcal{H}$ is the hypothesis space. If the hypothesis generation cost is $o(T)$, the dominant term scales as $\sqrt{\Gamma \log |\mathcal{H}|}\sqrt{T}$, yielding $O(\sqrt{T})$ convergence. This means the identification cost scales with the size of the hypothesis space through $\log |\mathcal{H}|$.

*Table 10.* **Theoretical comparison of learning efficiency: PIEVO vs. PiFlow.** Our PIEVO is theoretically more efficient than PiFlow. Though they share the same sublinear regret order, PIEVO enjoys a smaller complexity driver ($H(P)$) in the regret bound, as also empirically validated in Figure 7.

| Feature | PiFlow | PIEVO (Ours) |
|---|---|---|
| Learning space | $\mathcal{H}$ | $\mathcal{P}$ |
| Regret upper bound | $\mathcal{O}\left( \sqrt{T \cdot \log |\mathcal{H}|} \right)$ | $\mathcal{O}\left( \sqrt{T \cdot H(P)} \right)$ |
| Complexity driver | $\log |\mathcal{H}|$ ($\uparrow$) | $H(P)$ ($\downarrow$) |
| Convergence speed | Slower | Faster |

**The natural relationship of $|\mathcal{P}| < |\mathcal{H}|$ highlights the superior efficiency of PIEVO.** The efficiency gain of PIEVO hinges on the relationship $|\mathcal{P}| < |\mathcal{H}|$. Formally, let $\mathcal{H}$ be the universal set of hypotheses. A principle $P \in \mathcal{P}$ acts as a generator for a subset $\mathcal{H}_P \subseteq \mathcal{H}$. For the framework to be non-trivial, two conditions must hold (a) **coverage**, $\mathcal{H} = \bigcup_{P \in \mathcal{P}} \mathcal{H}_P$, and (b) **abstraction**, the average number of hypotheses per principle, $\bar{k} = \frac{1}{|\mathcal{P}|} \sum |\mathcal{H}_P|$, must be much greater than 1. By the union bound, $|\mathcal{H}| \leq \sum |\mathcal{H}_P| = |\mathcal{P}| \cdot \bar{k}$, which implies $|\mathcal{P}| \geq |\mathcal{H}|/\bar{k}$. For domains where principles offer strong abstraction, this directly leads to $|\mathcal{P}| < |\mathcal{H}|$, and consequently $\boldsymbol{H(P) \lesssim \log |\mathcal{P}| < \log |\mathcal{H}|}$. The first comparison uses the fact that $H(P)$ is the entropy of the principle random variable, while $\log |\mathcal{P}|$ is its cardinality-based upper bound.

> **Takeaway.** `PiFlow` lacks the dynamic principle evolution and IDS-driven selection, often operating in a space where the constant factor in regret is governed by the full hypothesis space through $\log |\mathcal{H}|$. In contrast, PIEVO shifts the learning complexity to the principle random variable $P$, whose information budget is quantified by $H(P)$.

## D. Deduction of Adopting the Information-Directed Sampling for Hypothesis Selection

We formulate scientific discovery as a dual-uncertainty problem. By decomposing the instantaneous regret $\Delta_t$, we identify that the primary theoretical bottleneck for achieving convergence is bounding the cumulative identification regret, $\sum_{t=1}^{T} \Delta_t^{(ID)}$.

**To bound the cost of identifying the right principle,** We recognize that the universal principle space 3.2 has a fundamental property: a finite prior entropy $H(p_0)$. Every experimental outcome provides some information gain $I_t$ about the true principle, and the cumulative information gain is strictly bottlenecked by this finite prior entropy

$$\sum I_t \leq H(p_0).$$

**To guarantee that the identification regret stops growing,** we must strictly anchor the accumulation of regret to this finite information budget. In other words, we cannot afford exploratory actions that incur high regret but yield zero information. Therefore, we can algebraically construct the connection between the cumulative identification regret and the information gain by writing:

$$\sum_{t=1}^{T} \Delta_t^{(ID)} = \sum_{t=1}^{T} \left( \frac{\Delta_t^{(ID)}}{\sqrt{I_t}} \cdot \sqrt{I_t} \right)$$

By Cauchy-Schwarz inequality, we have:

$$\sum_{t=1}^{T} \Delta_t^{(ID)} \leq \sqrt{\sum_{t=1}^{T} \frac{(\Delta_t^{(ID)})^2}{I_t}} \cdot \sqrt{\sum_{t=1}^{T} I_t}$$

As established, the total information gain is bounded:

$$\sum I_t \leq H(p_0)$$

Therefore, the inequality becomes:

$$\sum_{t=1}^{T} \Delta_t^{(ID)} \leq \sqrt{\sum_{t=1}^{T} \frac{(\Delta_t^{(ID)})^2}{I_t}} \cdot \sqrt{H(p_0)}$$

If the system ensures that

$$\frac{(\Delta_t^{(ID)})^2}{I_t} \leq \Gamma$$

at each step, then

$$\sum_{t=1}^{T} \frac{(\Delta_t^{(ID)})^2}{I_t} \leq \Gamma T.$$

Substituting this into the equation above, we yield our sublinear bound:

$$\sum_{t=1}^{T} \Delta_t^{(ID)} \leq \sqrt{\Gamma T} \cdot \sqrt{H(p_0)} = \mathcal{O}(\sqrt{T})$$

This imposes a strict mathematical design constraint: to guarantee sublinear convergence ($\mathcal{O}(\sqrt{T})$), the system's hypothesis-selection mechanism must explicitly constrain the ratio of squared regret to information gain ($\frac{(\Delta_t^{(ID)})^2}{I_t}$) at each step.

We seek a strategy that explicitly operationalizes this mathematically required regret-to-information trade-off. IDS (Russo & Van Roy, 2014) is precisely formulated to minimize this exact ratio. Therefore, the architecture of PIEVO naturally necessitates the adoption of IDS, over standard UCB or greedy generation, because its objective function structurally guarantees the algebraic prerequisite needed to exploit the bounded entropy of the principle space.

## E. Algorithm of PɪEᴠᴏ

While Section 3 establishes the theoretical foundations, the operational implementation of PɪEᴠᴏ follows a multi-agent coordination flow that allows for dynamic principle optimization. The typical execution loop is summarized in the following Algorithm 1.

**Operational Workflow of PiEvo.** The execution of PɪEᴠᴏ is structured as a closed-loop, multi-agent coordination process that operationalizes the evolution of principles through a Bayesian framework. As formalised in Algorithm 1, the system cyclically transitions through four primary phases: (1) **anomaly-driven principle generation**, (2) **posterior belief updating**, (3) **strategic hypothesis selection** via Information Directed Sampling (IDS), and (4) **empirical observation** by experimentation. This iterative loop ensures that the agentic system does not merely optimize within a fixed conceptual space but actively expands its foundational logic to account for unexpected environmental feedback.

## F. LLM Test-Time Reasoning Ablation

We assess the impact of inference-time reasoning (Gemini-2.5-flash `Think`) on discovery performance (Table 11). We observe a distinct dichotomy: unstructured baselines like `AI Researcher` benefit from internal chain-of-thought (↑ 25.06% SQ), as it compensates for their lack of strategic planning. Conversely, structured frameworks like `PiFlow` and PɪEᴠᴏ suffer performance degradation (↓ 16.78% and ↓ 26.35% SQ, respectively). This suggests an *interference effect*: effectively, the LLM's stochastic internal reasoning conflicts with the rigorous, data-driven utility functions (e.g., GP experts) of the higher-level framework, overriding optimal exploration decisions with sub-optimal heuristics. Thus, for principle-guided systems, externalized Bayesian reasoning proves superior to internal LLM "thinking".

*Table 11.* Comparison of Think and **No-Think** modes on MBO with `Gemini-2.5-flash` model. Arrows indicate performance improvement (↑) or degradation (↓) in Think mode relative to No-Think. All changes are highlighted as superior or inferior.

| Method | Mode | SQ (%) ↑ | AUOC (%) ↑ | APD (%) ↑ |
|---|---|---|---|---|
| AI Researcher | No-Think | $61.22 \pm 21.80$ | $49.13 \pm 9.99$ | $6.90 \pm 8.17$ |
| | Think | $76.57 \pm 28.20$ ↑25.06% | $52.42 \pm 7.64$ ↑6.71% | $14.09 \pm 8.78$ ↑104.16% |
| PiFlow | No-Think | $80.18 \pm 7.55$ | $52.81 \pm 6.05$ | $14.36 \pm 8.90$ |
| | Think | $66.72 \pm 2.37$ ↓16.78% | $46.78 \pm 3.33$ ↓11.41% | $13.78 \pm 10.52$ ↓4.06% |
| **PɪEᴠᴏ (ours)** | No-Think | $153.53 \pm 6.35$ | $122.26 \pm 9.14$ | $28.30 \pm 2.48$ |
| | Think | $113.08 \pm 21.71$ ↓26.35% | $94.48 \pm 11.57$ ↓22.72% | $22.54 \pm 4.07$ ↓20.33% |

## G. Internal Principle Evolving Dynamics of PɪEᴠᴏ

The high performance of PɪEᴠᴏ is fundamentally rooted in its ability to navigate the discovery process via a dual-layered uncertainty minimization framework. By this design, PɪEᴠᴏ achieves a level of sample efficiency that exceeds traditional methods. In this section, we provide a deep quantitative and qualitative analysis of the internal dynamics that drive this evolutionary process. All results are sampled from one running case with superlong running horizon (47 experiments in total).

**Evolutionary principle identification.** The core of PɪEᴠᴏ's cognition is the maintenance of a posterior distribution over a dynamically expanding principle space. As shown in Figure 9a, the belief mass initially disperses across multiple candidate principles. However, as experimental evidence accumulates, we observe a distinct concentration of belief on specific principles that exhibit high predictive accuracy.

A critical milestone in this process is the Watershed Phenomenon, visualized in Figure 10a. In scientific discovery, initial priors (often limited by pre-existing knowledge or LLM hallucinations) can frequently mislead the agent. The watershed represents the inflection point where the log-likelihood of accumulated outcomes provides sufficient evidence to overcome these prior biases. Beyond this point, the ground-truth principle $P^\star$ emerges as the dominant paradigm, drastically reducing $U^{\text{EP}}$ and enabling the agent to transition from broad exploration to targeted exploitation.

**Information-Directed execution efficiency.** In PɪEᴠᴏ, the mechanism for selecting experimental hypotheses is governed by Information-Directed Sampling (IDS), which balances the expected reward (exploitation) against the expected entropy

---

**Algorithm 1** Iterative Principle-Hypothesis-Testing Loop in PIEVO.

---

1: **Initialize:** Task and objective $\mathcal{O}$, Principle Agent $\mathcal{A}_P$, Hypothesis Agent $\mathcal{A}_H$, and Experiment Agent $\mathcal{A}_E$. Principle working set $\mathcal{P}_0$, uniform priors of principles $p_0(P)$, History of observations $\mathcal{H}_0 = \emptyset$, Gaussian Process Models $\mathcal{M}_P$ for all $P \in \mathcal{P}_0$ using RBF kernel, and fixed budget $T$.

2: **for** $t = 1, 2, \ldots, T$ **do**

3:     **1. Anomaly detection & Coherent augmentation (for Principle Agent $\mathcal{A}_P$)**

4:     Identify MAP principle: $P_{\text{MAP}} \leftarrow \arg\max_P p_{t-1}(P)$

5:     **for** each $(h_s, y_s) \in \mathcal{H}_{t-1}$ **do**

6:         Extract features $\boldsymbol{x} \leftarrow \phi(h_s, P_{\text{MAP}})$

7:         Predict $\mu, \sigma^2 \leftarrow \mathcal{M}_{P_{\text{MAP}}}(\boldsymbol{x})$

8:         Compute surprisal by $S_s = 1 - \exp\left(-\sqrt{\frac{(y_s - \mu_{\text{MAP}}(h_s))^2}{\sigma_{\text{MAP}}^2(h_s) + \sigma_{\text{obs}}^2}}\right) > \theta_t$ (by Equation 3)

9:     **end for**

10:     **if** $\max(S) > \tau_{adaptive}$ **then**

11:         $\mathcal{A}_P$ generates $P_{new}$ via LLM targeting high-surprisal anomalies

12:         $\mathcal{P}_t \leftarrow \mathcal{P}_{t-1} \cup \{P_{new}\}$, assign prior $p_0(P_{new})$

13:         **Back-fill:** Initialize $\mathcal{M}_{P_{new}}$ and train on **all** $(h, y) \in \mathcal{H}_{t-1}$

14:     **else**

15:         $\mathcal{A}_P$ generates $P_t$ for exploration or exploitation, depends on $p(P_{\text{MAP}})$

16:     **end if**

17:

18:     **2. Full-history posterior update (for System State Updating)**

19:     **for** each principle $P \in \mathcal{P}_t$ **do**

20:         $\log \tilde{p}(P) \leftarrow \log p_0(P)$

21:         **for** each observation $(h_s, y_s) \in \mathcal{H}_{t-1}$ **do**

22:             $\mu_s, \sigma_s^2 \leftarrow \mathcal{M}_P(\phi(h_s, P))$

23:             $\mathcal{L} \leftarrow \mathcal{N}(y_s; \mu_s, \sigma_s^2 + \sigma_{noise}^2)$

24:             $\log \tilde{p}(P) \leftarrow \log \tilde{p}(P) + \log \mathcal{L}$

25:         **end for**

26:     **end for**

27:     Normalize posteriors $p_t(P) \propto \exp(\log \tilde{p}(P))$

28:

29:     **3. Hypothesis selection by IDS (for Hypothesis Agent $\mathcal{A}_H$)**

30:     **if** Warm-up Phase **then**

31:         Select $h_t \leftarrow \arg\max_h \sum_P \sigma_P^2(\phi(h, P))$

32:     **else**

33:         **for** candidate $h$ **do**

34:             Estimate regret by $\Delta_t(h) \leftarrow \mathbb{E}_{p_t}[v^*] - \mathbb{E}_{p_t}[r(h)]$

35:             Estimate info gain $I_t(h)$ via Monte Carlo (BALD):

36:                 Sample $y^{(m)} \sim p(y|h)$, compute expected entropy drop $H(P) - \mathbb{E}[H(P|y^{(m)})]$

37:             Compute IDS ratio $\Psi_t(h) \leftarrow \Delta_t(h)^2 / (I_t(h) + \epsilon)$ (by Equation 1)

38:         **end for**

39:         Select $h_t \leftarrow \arg\min_h \Psi_t(h)$

40:     **end if**

41:

42:     **4. Observation of $y_t$ & Model update (for Experiment Agent $\mathcal{A}_E$)**

43:     Execute $h_t$, observe outcome $y_t$

44:     $\mathcal{H}_t \leftarrow \mathcal{H}_{t-1} \cup \{(h_t, y_t)\}$

45:     **for** each principle $P \in \mathcal{P}_t$ **do**

46:         Extract structural features $\boldsymbol{x}_t \leftarrow \phi(h_t, P)$

47:         Update GP posterior $\mathcal{M}_P$ with new pair $(\boldsymbol{x}_t, y_t)$

48:     **end for**

49: **end for**

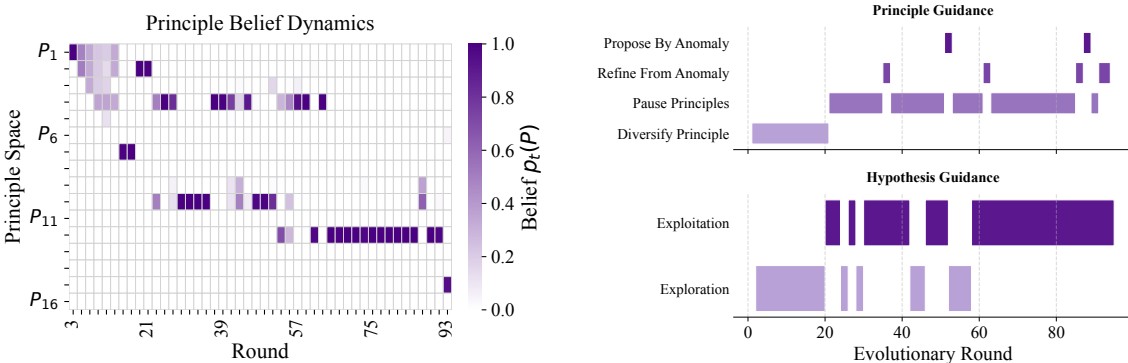

*(a)* **Belief Heatmap of Principles in Long Horizon.** *(b)* **Guidance Type of Principle Agent and Hypothesis Agent.**

*Figure 9.* **Evolutionary identification and guidance shifts in PIEVO.** (a) Heatmap of posterior probabilities $p_t$ at round $t$ over the expanding principle space, illustrating the concentration of belief on the True Principle $P^\star$. (b) Evolution of guidance types (see Appendix K), showing the transition from initial exploration to principle-guided exploitation. PIEVO dynamically changes the guidance type to fit the updating of principle beliefs, thereby making high quality hypotheses iteratively.

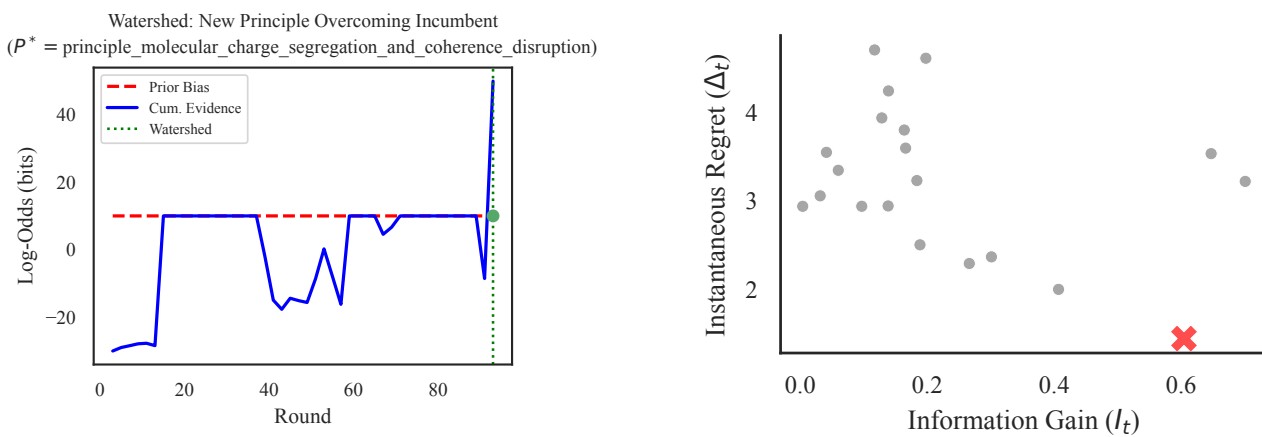

*(a)* **Watershed Phenomenon of Final Optimal Principle.** *(b)* **Sampled Information-Directed Hypothesis Selection.**

*Figure 10.* **Information-directed sampling and learning efficiency.** (a) The Watershed Phenomenon demonstrates the moment when accumulated experimental evidence overcomes the initial prior bias, triggering a transition from incumbent theories to the ground truth. (c) Visualization of a sampled IDS candidate space, where hypotheses are selected by minimizing the ratio of squared regret to information gain ($\Psi_t = \Delta_t^2 / I_t$), gray dots are accumulated hypothesis candidates, while the red icon is the selected one.

reduction in the principle space (exploration). Figure 10b illustrates the candidate space at a sampled iteration (Round 45). The agent evaluates potential hypotheses by minimizing the ratio $\Psi_t(h) = \Delta_t(h)^2 / I_t(h)$, effectively selecting actions that provide the highest information gain per unit of regret cost, as shown Figure 11.

> **Takeaway (Principle Evolutionary Dynamics).** The efficacy of PIEVO stems from the coupling of principle evolution and hypothesis selection. Across 45 NHO iterations, PIEVO successfully evolves principles to uncover high-quality candidates, while GP experts maintain calibrated likelihoods to drive the evolutionary dynamics.

## H. Gaussian Process Experts Analysis

The robust performance of PIEVO is fundamentally enabled by the predictive fidelity of its Gaussian Process (GP) experts. In our framework, each principle $P$ is operationalized through an associated GP expert $\mathcal{M}_P$. These experts serve as the bridge between qualitative semantic logic and quantitative experimental data, mapping the structural feature embedding $\phi(h, P)$ (see Section B) to the observed reward space. This section provides a quantitative evaluation of the GP experts' learning dynamics and their predictive accuracy.

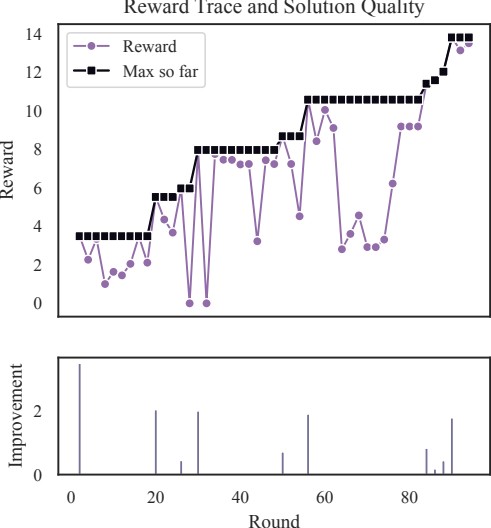

*Figure 11.* **Empirical solution quality and discovery rate.** The trajectory of observed outcome (reward) and cumulative best outcome reflects the system's ability to consistently improve solution quality as principle identification matures.

**Hyperparameter evolution and task adaptation.** A critical requirement for the GP experts is the ability to adapt their prior beliefs to the specific physical characteristics of the task. Figure 12a illustrates the dynamic evolution of the GP hyperparameters for the optimal principle identified during the discovery process. We observe a clear convergence in the hyperparameter space as the number of experimental observations increases. Specifically, the dynamic adjustment of the RBF kernel's lengthscales and signal variance indicates that the model is actively learning the "sensitivity" of the principle to different semantic dimensions. This online adaptation is crucial for uncertainty calibration; it ensures that the predictive uncertainty $\sigma_t(h)$ remains an accurate proxy for surprisal, thereby facilitating the efficient information gain pursued by the IDS objective.

**Regression accuracy and design validation of GP expert.** The effectiveness of the naive geometric features proposed in Section B is validated through the prediction performance of the GP experts. Figure 12b displays a parity plot comparing the predicted outcomes from the GP expert of the MAP principle against the ground-truth experimental results. Over a long-horizon discovery run, the GP model achieves a coefficient of determination ($R^2$) of 0.823, demonstrating that the semantic alignment between hypotheses and principles constitutes a sufficient and informative statistic for the experimental likelihood.

This empirical evidence confirms the theoretical hypothesis that scientific truth can be captured by the geometric relationship between a principle and its corresponding hypotheses, as we discussed in Section B. Furthermore, this predictive fidelity is the underlying driver of the Watershed Phenomenon observed in Figure 10; because the GP expert can reliably distinguish between models that explain the data and those that do not, the system can rapidly converge to an optimal paradigm, drastically reducing the cumulative regret while maximizing discovery potential.

> **Takeaway.** The GP experts transform the soft reasoning of Large Language Models (LLMs) into hard statistical constraints. The results confirm that PIEVO successfully integrates principle with hypothesis and experimental outcomes, allowing it to navigate complex discovery landscapes with a sample efficiency that traditional purely-reasoning-based methods cannot match.

## I. Case Study of PIEVO in the Discovery of Nanohelix Chiral Mechanism

To evaluate the autonomous discovery potential of PIEVO, we deploy it to address a specific challenge in nanophotonics: *identifying the geometric conditions for maximizing optical chirality in nanohelices by considering the electromagnetic interactions.*

As previous literature reports, the chiral response in a nanohelix array is governed by the interference between the toroidal

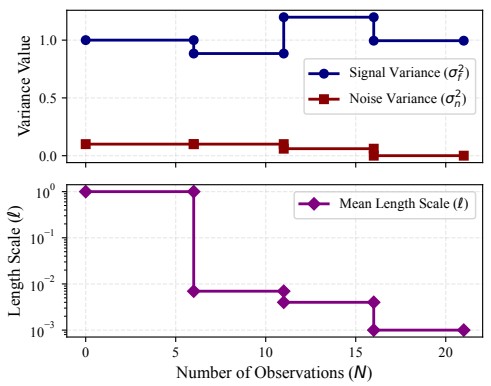
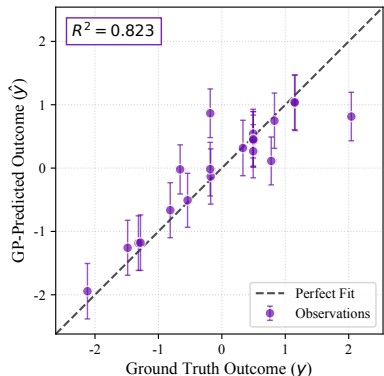

*(a)* **Hyperparameter Dynamics of Gaussian Process model.**   *(b)* **Sampled Performance of Gaussian Process model.**

*Figure 12.* **Gaussian Process expert analysis.** (a) Dynamics of GP hyperparameters demonstrating the adaptation to task-specific observation patterns. (b) Predictive accuracy of the optimal principle's GP expert, showing high fidelity ($R^2 = 0.823$) against experimental outcomes.

dipole ($T$) and the electric quadrupole ($Q_e$) ([Faniayeu et al., 2020](#)). However, it is unclear that in a single nanohelix system, the chiral response is governed by the same physics, because generally the phenomenon observed in array systems are not directly transferable to single nanohelix systems.

> **Task objective:** Maximize the optical chirality (g-factor) of a gold nanohelix by considering electromagnetic interactions.
> **Tunable parameters:** Fiber radius ($R_f$), helix radius ($R_h$), pitch ($Pitch$), turns ($N$), optics bi-direction (forward or backward), observation axis (x axis, y axis or z axis), and wavelength ($\lambda$).

**Scientific Challenges.**   This task presents two challenges for autonomous discovery, as detailed below:

1. First, the search space is **high-dimensional**, where the g-factor response is highly non-linear with sharp, narrow-band resonances.

2. Second, the agent must discern how the **circulation of solenoidal currents on the helical meridians** gives rise to a non-vanishing toroidal moment that supplements or even dominates the traditional dipole transitions.

### I.1. Trajectory and Analysis

The agent's exploration spanned multiple iterations of hypothesis generation, experimental validation, and principle refinement. We categorize this trajectory into four distinct phases:

**Phase 1: Initial mechanistic probing (round 1).** In the early rounds, the agent successfully identifies the **Toroidal anapole resonance** as a primary candidate. This leads to an initial high g-factor of 1.806 (about 90% SQ) for a tightly wound, thin-wire geometry ($P/R \approx 10$, $R = 20\,\text{nm}$).

**Phase 2: Divergent search and scalability testing (rounds 2-4).** During middle rounds, the agent intentionally diversified its principle space to avoid local minima, proposing mechanisms such as **Chiral Thermal Fluctuation Amplification** and **Quantum Casimir Mode Selection**. While these rounds yield lower g-factors (e.g., 0.450 and 0.923), they provide critical boundary data, establishing that chirality persists even in sparse or non-resonant configurations, though at reduced magnitudes.

**Phase 3: Anomaly-driven refinement (rounds 5-6).** A pivotal shift occurred when the agent encounters high g-factors in geometries that violate the strict "integer turn count" requirement of its Geometric Phase models. For instance, a non-integer turn count ($n\_turns = 9.8$) with compressed pitch ($P = 61\,\text{nm}$) yields a substantial $g \approx 1.64$. This anomaly forces the agent to refine its principles, leading to the discovery of the **Helix Angle Gate** ($\theta \in [45°, 70°]$) as a universal moderator of radiative efficiency.

**Phase 4: Synthesis and final convergence (rounds 7-8).** In the final stages, the agent unified its findings into a comprehensive framework. It recognized that maximal chirality emerge from the intersection of resonant mode excitation (geometric phase or toroidal interference) and optimal current-to-radiation impedance matching (electrical thinness relative to skin depth). This led to the final g-factor optimization toward 1.83.

## I.2. The Evolution of Principles

Table 12 summarizes the iterative evolution of the scientific principles proposed by the agent and the corresponding experimental outcomes (with maximum of 2.0 theoretically).

*Table 12.* Evolution of discovered principles and experimental outcomes in the Nanohelix Optimization.

| Round | Proposed Principle | Best $g$-factor | Mechanism Class |
|---|---|---|---|
| 1 | Toroidal Anapole Resonance | 1.806 | Multipolar Interference |
| 2 | SPP Bragg Scattering | 1.617 | Plasmonic |
| 4 | Thermal Fluctuation Amplification | 1.518 | Stochastic |
| 11 | Geometric Phase Refined (V1) | 1.647 | Geometric Phase |
| 15 | Chiral Impedance Mismatch | 1.717 | Radiative Coupling |
| 18 | Multipolar Phase Matching | 1.796 | Hybrid |
| 20 | **Final Unified Principle** | **1.826** | Multi-Gated Resonance |

## I.3. The Final Scientific Principle Produced by PIEVO

The agent's final submission represents a sophisticated synthesis of electromagnetic theory and structural optimization. We reproduce the agent's verbatim conclusion below:

> *"Maximum g-factor in metallic nanohelices arises from either (a) **geometric-phase-mediated spin–orbit conversion** (tight helix_radius $\leq 25$ nm, integer or near-integer turns, long pitch $\geq 190$ nm) or (b) **toroidal-electric quadrupole interference** (pitch $< 90$ nm, high turn density)—but **only when two gating conditions are simultaneously satisfied**: (i) the helix angle $\theta = \arctan(pitch/(2\pi \cdot helix\_radius))$ lies between $45°$ and $70°$ **or**, if $\theta$ is suboptimal ($30°$–$45°$), the axial length exceeds the wavelength to enable longitudinal phase accumulation; and (ii) the fiber_radius is less than or comparable to the skin depth at the operating wavelength (fiber_radius $\leq \delta_{skin}(\lambda)$), ensuring surface-confined, phase-coherent conduction electron oscillations. When both conditions hold, resonant multipolar content couples efficiently to polarization-asymmetric radiation channels; when either fails, g-factor is suppressed regardless of geometric winding or P/R ratio. This refinement unifies observed anomalies by recognizing that chirality requires not just the right shape, but the right electromagnetic 'wiring'—thin enough to avoid internal phase decoherence and angled enough to radiate asymmetrically."*

This principle accounts for the failure of large-radius or thick-wire designs while justifying the peak performance of elongated, sub-skin-depth helices.

## I.4. Manual Post-Validation via FDTD Simulation

To confirm that the agent's evolving principles reflect genuine physical phenomena rather than statistical coincidences, we performed rigorous full-wave electromagnetic simulations using the Finite-Difference Time-Domain (FDTD) method as the ground truth, as shown in Figure 13. We focused on the two primary mechanisms identified by PIEVO: (a) the **geometric phase** gradient resonance the **multipolar resonant** state. As the geometric phase can be directly validated by the surrogate model, our simulation primarily focuses on the multipolar resonant state.

**Evidence 1: Multipolar spectral convergence.** We simulate the optimized short-pitch geometry ($P = 60$ nm, $R_h = 25$ nm) across the visible spectrum. As shown in Fig. 13a, the scattering decomposition reveals a dominant excitation of the toroidal dipole ($T$). Crucially, at the agent's predicted resonance ($\lambda \approx 450$ nm), the total effective dipole $|P + ikT|^2$ is strongly suppressed due to destructive interference (the "Anapole" condition), while the magnetic dipole and electric quadrupole remain active. **Following electrodynamic theory, this asymmetric suppression for LCP versus RCP light is the fundamental driver of the giant g-factor found by the agent.**

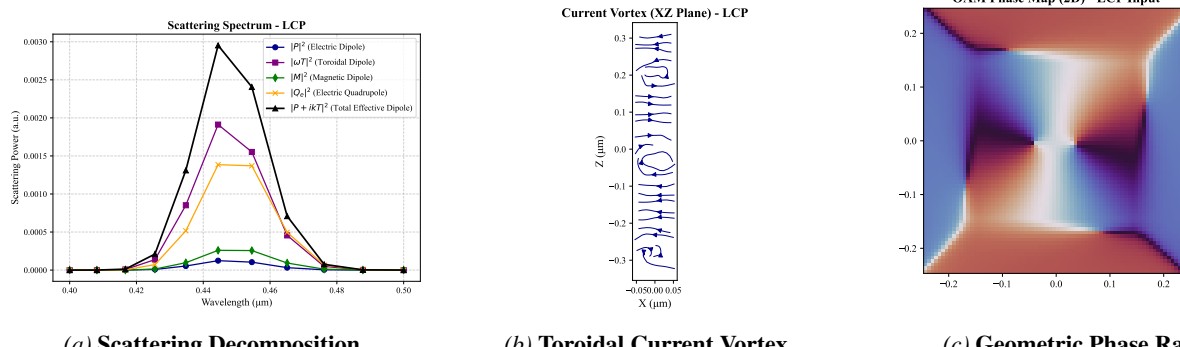

*(a)* **Scattering Decomposition.**   *(b)* **Toroidal Current Vortex.**   *(c)* **Geometric Phase Ramp.**

*Figure 13.* **Full-wave FDTD verification of the agent's discovered principles.** (a) Spectral suppression of the effective dipole moment at the anapole resonance. (b) Visualization of the poloidal current loop supporting the toroidal dipole. (c) Transmitted phase distribution confirming geometric phase-induced wave-front shaping.

**Evidence 2: Current topology and phase landscapes (Figure 13b and Figure 13c).**   To verify the internal physics, we extract the current density distribution and transmitted phase ramps, and there are two observations:

1. As shown in Fig. 13b, the streamplot of the current density in the $XZ$-plane exhibits a distinct poloidal vortex, which is the singular signature of a toroidal dipole.

2. As shown in Fig. 13c, for the long-pitch regime ($P \approx 200\,\mathrm{nm}$), the $XY$-plane phase distribution of the cross-polarized transmitted field shows a spiral phase ramp corresponding to a topological charge of $l = 1$. This is the direct evidence of the strong chirality powered by toroidal moment.

This confirms the hypothesis from agent that the nanohelix acts as a geometric phase-gradient resonator, imparting polarization-dependent orbital angular momentum (OAM) to the light.

**Remark (Discovery vs. Retrieval).** While the electrodynamic theory is fundamental and has been well-studied in optics area, the success of PIEVO in identifying mechanisms in the unknown system suggests high potential in aiding the scientific discovery. Importantly, initial rounds of the agent frequently proposed generic or slightly incorrect mechanisms (e.g., simple SPP Bragg scattering). It was only through the **iteration-anomaly-refinement** cycle, specifically after encountering high g-factors in electrically thin fibers and at non-integer turn counts, that the agent formulated the highly specific *Helix Angle Gate* and *Skin Depth Scaling* constraints.

---

**Takeaway.** In this study, PIEVO discovers principles, i.e., structural-property correlations specific to the metallic toroidal regime that, are not present in standard textbook descriptions of nanohelices. Thus, the performance of PIEVO represents a genuine *de novo* scientific discovery driven by evidence rather than mere pattern matching.

---

## J. Benchmark and Task Definitions

We follow Pu et al. (2025) to use high-fidelity surrogate models for agents to execute hypothesis and obtain external feedback, with the same theoretical maximum or reference value, $\mu_{ref}$, for computing metrics (see Section 4.2): (a) for NHO task, $\mu_{ref} = 2.0$ as the theoretical maximum of optical chirality, (b) for MBO task, we reference the empirical value $\mu_{ref} = 6.5$ from Pu et al. (2025), (c) for SPO task, we use the reference value of $\mu_{ref} = 298.5$, which is equal to room temperature.

Similarly, we use the same scenario of nanohelix material optimization (NHO), bio-molecular optimization (MBO) and superconductor critical temperature optimization (SPO). Additionally, we follow Song et al. (2025) to use transition metal complexity optimization (TMC) task with $\mu_{ref} = 493.8$ to further evaluate our method. In subsections below, we discuss the detailed configurations of each task.

## J.1. Nanohelix Optimization (NHO)

Synthesizing materials often need to consider multiple factors, not just structure parameters of material itself, but also the environment conditions that lead to the target observation.

As reported, the NHO task in (Pu et al., 2025) only uses 4 parameters, i.e., helix radius, fiber radius, the number of turns and pitch length. Though these parameters can fully describe a nanohelix material, however, the target objective of chirality, essentially depends on the wavelength, optical direction, etc. Thus, we expand the scope of optimizing its geometric structure into the optimization of environment additionally, allowing higher dimensional hypothesis space than NHO task in PiFlow.

In our evaluation, the NHO task now includes: *helix radius, fiber radius, the number of turns, pitch length, direction of light (forward or backward), observation axis (x axis, y axis or z axis) and the wavelength*. We train the surrogate model based on the dataset from Pu et al. (2025) with same training protocol, and obtain the high-fidelity prediction model, the overall accuracy is up to 99.36%, as shown in Figure 14. In our experiments, **we run all baselines on this seven-dimensional NHO task for a fair comparison.**

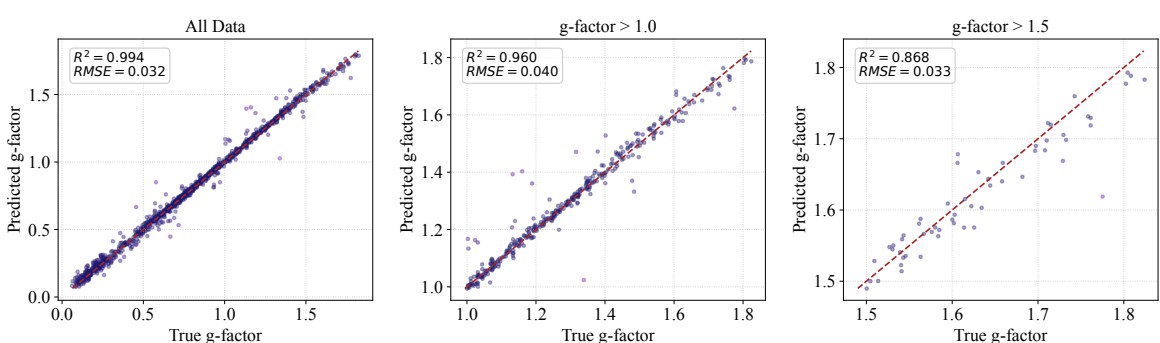

*Figure 14.* **Performance of Surrogate Model in NHO Task.**

**Task and Objective.** In NHO task, SQ $\sim 100\%$ means the optimized nanohelix structure is approximately absorbing all light from the incident direction, indicating strong chirality. We define the exact task description below:

---

**Nanohelix Structure Optimization**
Find the parameter combination that maximizes its optical chirality (described by $g$-factor value $\in [0, 2]$).

**[Parameters for Hypothesizing (Important)]**
- **fiber-radius** (20–60 nm): Radius of the actual fiber/wire that forms the helix (nm), or simply the thickness of a helix's wire. If the wire is 40nm thick, `fiber_radius=20`.
- **helix-radius** (20–90 nm): The distance from central axis to the center of the helical path (nm). Simply, this is the radius of the cylinder that a spring wraps around.
- **n-turns** (float, 3.0–10.0): The number of turns in the helix.
- **pitch length** (60–200 nm): Axial distance between adjacent turns or vertical spacing between turns.
- **wavelength** (float, 400.0–800.0 nm): The specific optical wavelength of the incident light at which the $g$-factor is evaluated. The highest chirality often occurs in the 400–500 nm range.
- **x_y_z** (integer, 0, 1, or 2): Specifies the principal coordinate axis for the light source: 0 represents the $x$-axis, 1 represents the $y$-axis, and 2 represents the $z$-axis.
- **direction** (integer, 0 or 1): Indicates the polarity of light propagation along the selected axis: 0 represents the Positive (Forward) direction, and 1 represents the Negative (Backward) direction.

**[Target Property for Optimizing]**
- $g$-**factor** (range from 0.0 to 2.0) as high as possible, higher value means stronger chirality. This is always computed by `characterize_nanohelix_gfactor` tool of Experiment Agent.

**[Instruction of Hypothesis/Principle Proposal]**
- Do not introduce any numerical values in proposed principles.
- Describing the correlation or a dependency that can help to find the best chiral property with highest $g$-factor value is allowed.
- Every principle MUST predict a testable parameter-performance trend of parameter interactions.

---

> - Every hypothesis must include exact values of all four parameters: `fiber-radius`, `helix-radius`, `n-turns`, `pitch`, `wavelength`, `x_y_z`, and `forward_backward`.
> - Do not introduce complex or meta principles that are beyond the parameter level, e.g., the quantum or electronic effects.
>
>   **[Formation]**
> - Every proposed hypothesis MUST follow the form of a string, as exampled in the *Last Record of the Experiment* below.
> - Minor increase or decrease any parameters around $\pm 0.1$ is not allowed.
>
> ________________________________________________________________
>
>   **[Last Record of the Experiment]**
> - **Hypothesis** (string format): `"fiber_radius=20.0, helix_radius=60.0, n_turns=6.0, pitch=20.0, wavelength=400.0, x_y_z=2, forward_backward=0"` (A hypothesis MUST include all 7 parameters for doing experiment, any other forms or combinations will be strongly rejected)
> - **Chiral property value:** $g$-factor $= -0.2613$

**Remarks on Dimensionality and Optimal Bounds.** A counter-intuitive phenomenon illustrated in Table 3 is that PIEVO achieves higher Solution Quality (SQ) in the 7-dimensional task ($96.36\%$) compared to the lower-dimensional variants (e.g., 4D at $84.56\%$). While reducing dimensionality typically mitigates the *curse of dimensionality*, implying easier optimization, our ablation setup constructs lower-dimensional tasks by **fixing** specific environmental parameters (e.g., observation axis or light direction) to constant values. Physical analysis reveals that the optical chirality of nanohelices is highly anisotropic (Wu et al., 2025a). The global optimum ($g \approx 2.0$) resides in a narrow manifold requiring precise alignment of the incident light vector. In 4D, 5D, and 6D settings, the fixed parameters effectively **exclude** this global optimum from the search space, imposing a hard physical ceiling on the achievable g-factor ($\mu_{max}^{constrained} < \mu_{ref}$). Since SQ is consistently calculated against the global reference ($\mu_{ref} = 2.0$), the lower scores in restricted settings reflect this physical bound rather than a deficiency in search efficiency. Conversely, the 7-dimensional space, despite its volume, fully encompasses the global optimum, enabling PIEVO to exploit the complete parameter set to reach peak chirality.

## J.2. Molecular Bio-activity Optimization (MBO)

We keep the task as the same with PiFlow (Pu et al., 2025). The agents are tasked with the objective of searching the molecular structures that has highest bio-activity, quantified by its pChEMBL value.

## J.3. Superconductor Optimization (SPO)

In SPO task, we found that the searching space involves much more entities that may lead to reward hacking, i.e., leveraging the limitations of surrogate models to propose candidates with extreme values that is outside the training data scope. To avoid this issue, we constrain the searching scope of original SPO task in PiFlow (Pu et al., 2025) to specifically five superconductor systems. As shown in Figure 15, we plot the $T_c$ distribution of five representative superconductors. Our task is to let agents search among given material systems to reach the superconductor formula with the highest $T_c$.

**Task and Objective.** We define the exact task description below:

> This team work **MUST** follow the standard scientific research. This research task aims to discover a superconductor with a larger critical temperature (`Tc`, measured in Kelvin). All members operate within a Hypothesis-Validation mechanism. Our ultimate objective is to discover a material (within the given systems) with a `Tc` value as high as possible, approaching room temperature (298.15 K).
>
>   **[Important] Requirements:**
> The discovery process involves exploring the vast search space of elemental combinations and their specific atomic ratios by hypothesizing and testing its underlying physicochemical principles proposed by your group members. The element formula **MUSTN'T** consider any environment conditions, only *Element-first & Ratio-second* are allowed.
>
>   **[Important] Suggested Chemical Systems for Exploration**
> To accelerate discovery, we should **ONLY** focus on exploring candidates **within Copper Oxide chemical systems**:
> [Only considering Elements (limited in each system) and Ratios (integer or float)]
> - Available systems are listed for you to design formulas to discover higher `Tc` superconductors:
> - La-Sr-Cu-O System
> - La-Ba-Cu-O System
> - Y-Ba-Cu-O System

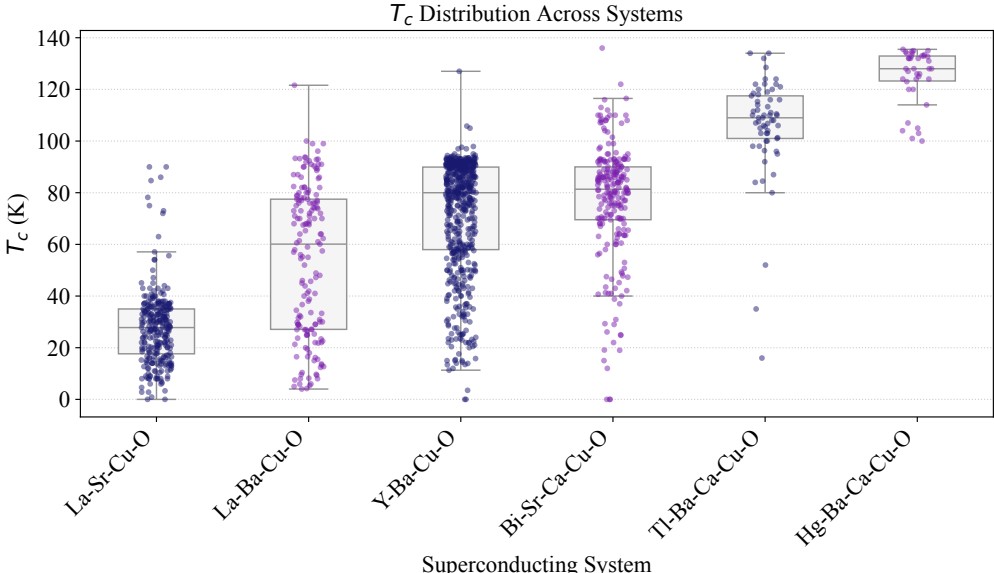

*Figure 15.* **Representative Systems of Superconductors Supported by Surrogate Model.**

---

- Bi-Sr-Ca-Cu-O System
- Tl-Ba-Ca-Cu-O System
- Hg-Ba-Ca-Cu-O System

**[EXPERIMENTAL REQUIREMENTS]**
- **Candidate Format:** The `Candidate` **MUST** be a single string representing the material's exact Stoichiometric Formula.
- **Formula Content:** The formula string **MUST** only contain valid chemical element symbols and their numerical ratios.
- **A Note on Ratios:** The experimental measurement is deterministic and based on atomic proportions.
- **[STRICT] DO NOT** add conditions like "xxx/Nb", "xxx-Layered", etc. Only element with ratios (float) are acceptable.

---

**[Example record of the experiment]**
- **Candidate:** `"Y1Ba2Cu2.7Co0.3O6.88"` (Must follow the given systems to make hypotheses)
- **Property (Tc):** 30.54 K

## J.4. Transition Metal Complex Optimization (TMC)

To further evaluate the performance of PiEvo in the domain of materials science, we introduce the Transition Metal Complex (TMC) Optimization task, adapted from the work of Song et al. (2025). This task challenges the agents to discover novel transition metal compounds with optimized properties, specifically focusing on the trade-off between structural complexity and material performance.

From the perspective of hypothesizing, this task requires agent to use exact 4 molecules to form a new compound, and the property of the new compound is evaluated by the surrogate model. Original task is to retrieve from a TMC database (Song et al., 2025). To let agent hypothesize divergently, we train a surrogate model by using all of its data corpus, including 1367499 records of given 50 ligands. Each TMC is constructed by four ligands in this pool.

As shown in Figure 16a, the surrogate model achieves an $R^2$ score of 1.00, indicating its capability to predict the properties of TMCs. Furthermore, we plot the distribution of TMCs in Figure 16b. We found that most of polarisability, i.e., the target property of TMC, are centered on the 200–300 range, and the distribution is approximately normal.

**Task and Objective.** We define the exact task description below:

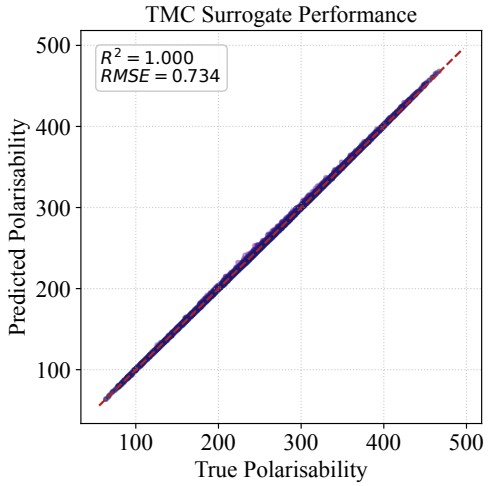

*(a)* **Accuracy of Surrogate Model.**

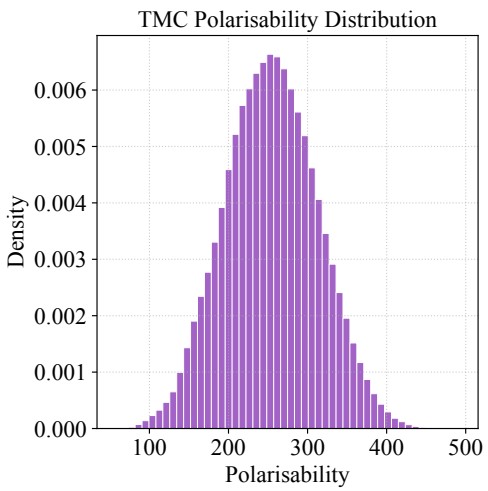

*(b)* **Distribution of TMCs.**

*Figure 16.* **TMC Surrogate Model Accuracy and Property Distribution.**

---

The goal is to design a chemically valid, charge-neutral Transition Metal Complex (TMC) by selecting exactly four ligands from a specific candidate pool to coordinate with a central Palladium cation ($Pd^{2+}$) to maximize the polarizability (the ability of the electron cloud to deform). This task requires you to follow principles of physical organic chemistry, e.g., Hard and Soft Acids and Bases (HSAB) Theory and Electron Delocalization Theory.

**1. Objective**
- **Goal:** Identify the specific combination of ligands that maximizes the polarizability value (theoretical maximum value is 500.0).
- **Target Property:** Electronic Polarizability. Higher values are better.
- **Chemical Context:**
    - **Center:** Palladium (II) ion ($Pd^{2+}$).
    - **Ligands:** You must select 4 ligands (SMILES) from the provided pool.
- **Guiding Principles:**
    1) Only focus on the molecular level interpretations for making hypotheses.
    2) Do NOT mention quantum theory or unrelated physics for this task.
    3) This task only involves molecular, atomic, and electronic properties, simply.
    4) You could discover the mechanisms mentioned above yourself.

**2. Critical Constraints**
Any candidate that violates the following rules will be considered a FAILED experiment (outcome = 0).
- **Charge Neutrality Rule (Strict):**
    - The Pd center has a charge of +2.
    - The total complex MUST be neutral (Charge = 0).
    - Therefore, the sum of charges of the 4 selected ligands MUST be -2.
- **Ligand Combination Logic:**
    - The provided ligand pool ONLY contains ligands with charge -1 or 0.
    - To satisfy the Charge Neutrality Rule ($-2$ total), you MUST select:
        • Exactly TWO (2) ligands with charge $-1$.
        • Exactly TWO (2) ligands with charge 0.
    - Do not attempt any other combination (e.g., four $-0.5$ charge ligands do not exist in the pool).
- **Pool Restriction:**
    - You are strictly limited to the specific ligands provided in the table below.
    - No external molecules are allowed.

**3. Output Format Requirements**
The team must output the hypothesis candidate string strictly adhering to this format:
- **Prefix:** Must start with `Pd_`.
- **Identifiers:** Use the exact SMILES string (e.g., `O=[C-]OC`) from the table. DO NOT use Formal Names.
- **Separator:** Use underscores '_' to connect the Pd and the 4 ligand SMILES strings.

*Table 13.* **Ligand Pool for TMC Optimization Task.** The pool is divided into anionic ligands (Charge = -1) and neutral ligands (Charge = 0). The TMC task requires agents to select two ligands from each category to form a TMC complex.

| 1. Ligands with Charge = -1 | | 2. Ligands with Charge = 0 | |
| --- | --- | --- | --- |
| SMILES | Charge | SMILES | Charge |
| S1(=O)(=O)[N-]C(=O)c2c1cccc2 | -1 | n1c(cc(cc1C)C)C | 0 |
| O=[C-]OC | -1 | n1ccc(cc1)C | 0 |
| C1=C[C-]=CC=C1 | -1 | [C-]#[N+]C(C)(C)C | 0 |
| [I-] | -1 | CP(C)c1ccccc1 | 0 |
| [S-]c1c(c(cc(c1F)F)F)F | -1 | n1cccc(c1)Cl | 0 |
| C1CC(=O)[N-]C1=O | -1 | [C-]#[N+]C1CCCCC1 | 0 |
| O=N(=O)[O-] | -1 | n1c(cccc1C)C | 0 |
| [C-]1=CC=C(C=C1)F | -1 | O | 0 |
| [CH3-] | -1 | N#CC | 0 |
| [C-]1=CC=C(C=C1)C | -1 | S(C)C | 0 |
| O=N[O-] | -1 | c1ccnc(c1)N | 0 |
| [C-](F)(F)F | -1 | n1ccn(c1)C | 0 |
| [Cl-] | -1 | CN1[C]N(C)C=C1 | 0 |
| [C-]#N | -1 | n1ccc(cc1)N(C)C | 0 |
| [N-]=[N+]=[N-] | -1 | n1[nH]c(cc1C)C | 0 |
| [Br-] | -1 | [C-]#[O+] | 0 |
| [S-]C#N | -1 | O1CCNCC1 | 0 |
| [N-]1C(=O)c2c(C1=O)cccc2 | -1 | NCC | 0 |
| [C-]1=C(F)C(=C(C(=C1F)F)F)F | -1 | CS(=O)C | 0 |
| c1c(C#[C-])cccc1 | -1 | N | 0 |
| [O-]c1ccccc1 | -1 | c1ccccn1 | 0 |
| [F-] | -1 | S(=O)(C)C | 0 |
| [S-]c1ccccc1 | -1 | [C-]#[N+]c1c(C)cccc1C | 0 |
| C[C-]=O | -1 | CP(C)C | 0 |
| | | C(C)NCC | 0 |

---

- **Format Template:** `Pd_{SMILES}_{SMILES}_{SMILES}_{SMILES}`
- **Example String:** `Pd_c1ccccn1_S(=O)(C)C_C1=C[C-]=CC=C1_[N-]=[N+]=[N-]`
- **The SMILES:** All used SMILES string MUST be in the given table; do not change styles or use equal format, otherwise the SMILES cannot be recognized.

**4. Limited Ligands Pool** (See Table 13)

- **HINT:** You could choose two from the IDs with charge=0, and other twos form the IDs with charge=−1 to meet with the constraint.
- **[Critical]** All charges of SMILES MUST be referenced from the table above; DO NOT only judge its charge from the SMILES string.

**5. Explanation of SMILES expression for molecules/ligands**
- **Explicit Charge Syntax vs. Name:** BiasCharge is only defined by $+$ or $-$ symbols inside square brackets (e.g., `[O-]`, `[N+]`, `[Fe+2]`). An agent must strictly parse these tokens and ignore biases from chemical names.
- **Aromaticity and Electron Delocalization:** Atoms represented by lowercase letters (e.g., c, n, s, o) denote participation in an aromatic $\pi$-system. This is crucial for maximizing polarizability, as these atoms contribute significantly to the delocalized electron cloud.
- **Strict Valency in Square Brackets:** Atoms enclosed in square brackets (e.g., `[nH]`, `[C]`, `[Si]`) have their properties explicitly defined. If a specific H-count or charge is not written inside the bracket, it is zero.
- **Branching Hierarchy via Parentheses:** Parentheses `()` denote a branch connecting to the immediately preceding atom.
- **Ring Closure Markers:** Numeric digits (e.g., 1, 2) appearing after atoms indicate ring connection points. The agent must pair identical numbers to "close" the loop.

---

**[Last Record for Optimizing the Polarizability of TMC]**
- **Hypothesis:** "`Pd_c1ccccn1_CP(C)C_[Cl-]_[Br-]`"
- **Tested Polarizability (outcome):** 198.45
- **Hypothesis:** "`Pd_c1ccccn1_S(=O)(C)C_C1=C[C-]=CC=C1_[N-]=[N+]=[N-]`"
- **Tested Polarizability (outcome):** 235.25

**[Task]** Based on the guiding principles and history, propose the next single best candidate string that is expected to yield the highest polarizability. Ensure it strictly follows the "2 negative + 2 neutral" combination rule.

# K. Prompt Templates in PIEVO

This section details the system prompts governing the three constituent agents in PIEVO. As delineated in Section 3, instructions are dynamically synthesized based on the evolving epistemic state—specifically, the posterior beliefs over the principle space $p_t(\mathcal{P})$ and detected anomalies $\mathcal{U}_t$. These templates calibrate agents to perform specialized cognitive operations: Principle Management ($\mathcal{A}_P$), Hypothesis Generation ($\mathcal{A}_H$), and Experimental Execution ($\mathcal{A}_E$). For readability, we present semantic instructions while omitting technical JSON schema constraints.

### K.1. Guidance for the Principle Agent

The Principle Agent regulates the evolution of the principle space $\mathcal{P}$, underpinning the *Coherent Principle Augmentation* mechanism. Based on the anomaly detection state (Equation 3), PIEVO selects one of the following instructions.

**Diversifying Principle Space.** During initialization or principle space collapse, the system directs the agent to propose orthogonal mechanisms to maximize conceptual coverage.

> **[DIVERSIFYING PRINCIPLE SPACE]**
> **EXISTING PRINCIPLES AND BELIEFS**
> `{Active Principles}`
> **OBJECTIVE: MAXIMIZE DIVERSITY**
> - Propose **one novel, creative principle** to address the current problem (target: high reward).
> - Formulate mechanisms that are **conceptually distant** from the existing set.
> - Each principle must be self-contained, incorporating its logical premises and reasoning context.
> - Incremental modifications or minor variants of existing principles are **strictly prohibited**.

**Anomaly-Driven Augmentation.** Systemic triggers for augmentation occur when high-surprisal observations contradict current beliefs. Instructions diverge based on the confidence of the Maximum A Posteriori (MAP) principle.

*(Scenario A) High Confidence with Local Anomalies:* The primary principle is valid but requires boundary refinement.

> **[HIGH CONFIDENCE, LOCAL ANOMALIES]**
> **CURRENT TOP-POTENTIAL PRINCIPLE:** "`{top_principle_text}`"
> This principle demonstrates performance but fails to account for the following anomalies:
> `{anomalies}`
> **CURRENT PRINCIPLE SET**
> `{Active Principles}`
> **OBJECTIVE: REFINE AND RECONCILE**
> - **Do NOT** replace the top principle. Refine it based on local evidence.
> - Propose a variant or refinement by synthesizing commonalities among anomalies.
> - Integrate specific exceptions or boundary conditions to resolve failures without compromising core validity.

*(Scenario B) Low Confidence with Systematic Failures:* Current principles are fundamentally insufficient, necessitating new conceptual discovery.

> **[LOW CONFIDENCE, SYSTEMATIC FAILURES]**
> Significant anomalies have been detected that current principles cannot explain:
> `{anomalies}`
> **CONTEXT: INSUFFICIENT PRINCIPLES**
> `{Active Principles}`
> **OBJECTIVE: CONCEPTUAL DISCOVERY**
> - Perform pattern analysis on these failures and bridge them with auxiliary knowledge.
> - Propose a **novel, creative principle** that resolves the underlying cause of these outcomes.
> - The new principle must maximize explanatory power across patterns missed by existing principles.
> - To ensure interpretability, the principle **MUST** align strictly with documented observations.

**Silence (No Action).** To prevent hallucination and redundancy, if no anomalies are detected and the principle set is stable, the agent is instructed to return nothing.

### K.2. Guidance for the Hypothesis Agent

The Hypothesis Agent is governed by the *Information-Directed Selection* (IDS) strategy, generating candidates to either exploit high-posterior principles or explore high-uncertainty regions.

**Exploitation Mode.** When a dominant principle ($P^\star$) emerges, the agent is directed to maximize outcomes through guided exploitation.

> **Task Context:** {_task}
>
> **EXPERIMENTAL MEMORY (TESTED HYPOTHESES)**
> {tested candidates}
>
> **TARGET SCIENTIFIC PRINCIPLE**
> "{top belief principle text}"
>
> **INSTRUCTION**
> Propose **three** novel candidate hypotheses in JSON format following the target principle. Given our high confidence in $P^\star$, your objective is to **maximize rewards** by rigorously applying this principle to derive superior candidates.
>
> **REQUIREMENTS**
> - Generate 3 distinct, untested hypotheses (A, B, C) with high predicted utility.
> - **STRICTLY PROHIBITED:** Do not propose any hypothesis already present in the memory list.
> - Utilize patterns from the exploitation memory to inform and differentiate your proposals.

**Exploration Mode.** During initialization or high principle uncertainty, the agent performs broad exploration to reduce entropy in the hypothesis space.

> **Task Context:** {_task}
>
> **EXPERIMENTAL MEMORY (TESTED HYPOTHESES)**
> {tested candidates}
>
> **INSTRUCTION**
> Provide exploratory candidate hypotheses in JSON format:
> {task description}

### K.3. Guidance for the Experiment Agent

The Experiment Agent serves as the execution layer. It is constrained to implement the specific hypothesis $h_t$ selected via IDS, ensuring high-fidelity translation from theoretical selection to empirical results.

> **PRIMARY TASK (EXPERIMENT EXECUTION)**
> {task}
>
> **INSTRUCTION**
> Execute the assigned hypothesis using the provided tools.
>
> **SELECTED HYPOTHESIS**
>
> - {selected hypothesis}
>
> **OPERATIONAL CONSTRAINTS**
> - You are **only** authorized to test the specified candidate; others are strictly forbidden.
> - Upon completion, submit results via the EXPERIMENT_SUBMISSION signal.
> - Ensure the EXPERIMENT_SUBMISSION token is correctly assigned following tool output.

