# OpenReview forum: "Principle-Evolvable Scientific Discovery via Uncertainty Minimization"
_ICML.cc/2026/Conference — ICML 2026 regular_

### Official Review · Reviewer_7yvo · 2026-03-07

**Soundness:** 3
**Presentation:** 3
**Significance:** 3
**Originality:** 3
**Overall Recommendation:** 4
**Confidence:** 2

**Summary:**

This paper proposes the PiEvo framework, which aims to improve LLM-based automated scientific discovery methods. Traditional approaches typically search within a fixed hypothesis space, making them constrained by initial prior knowledge and limiting their ability to discover phenomena beyond existing theories. The core idea of PiEvo is to formulate scientific discovery as Bayesian optimization over an evolvable space of scientific principles. By dynamically evolving principles rather than merely searching hypotheses, the framework improves exploration efficiency and discovery potential. The paper evaluates the method on four scientific discovery tasks. Experimental results show that PiEvo outperforms existing methods in terms of solution quality, convergence speed, and the balance between exploration and exploitation. In the Nanohelix case study, PiEvo autonomously discovered an electrodynamic mechanism, demonstrating its potential for tackling complex scientific problems.

**Compliance With Llm Reviewing Policy:**

Affirmed.

**Final Justification:**

The author has addressed my concerns. After reviewing the feedback from the other reviewers, I believe a score of 4 may be appropriate.

**Key Questions For Authors:**

1. PiEvo’s principle evolution relies on anomaly thresholds and high-surprise data. Are these parameters sensitive to the choice across different scientific domains and experimental noise conditions? Is there an automatic tuning mechanism to ensure robustness?
2. If smaller or medium-sized models are used, or under low-resource environments, how would the performance of PiEvo change? Are there feasible adaptation strategies?
3. In the NHO tasks, the dimensionality is only 4-6. As the dimensionality of the principle space or hypothesis space increases, how much would PiEvo’s computational cost and sample efficiency be affected? Would there be a decline in convergence speed?

**Limitations:**

yes

**Strengths And Weaknesses:**

**Strengths:**
1. PiEvo demonstrates good innovation in its approach to scientific discovery.
2. This paper includes detailed theoretical analyses, including convergence, cumulative regret bounds, and sample complexity.
3. Compared with the baselines, PiEvo shows a substantial performance improvement.

**Weaknesses:**
1. The performance of PiEvo highly depends on the generative capability of the LLM and the accuracy of probability predictions from GP experts.
2. The theoretical section assumes strict hierarchical sampling and global entropy management, but the actual implementation uses marginalized IDS and heuristic approximations. Although the authors claim that the two share the same objective, the theoretical guarantees may be relaxed in practice and may rely on implicit assumptions.
3. Although PiEvo performs well on NHO tasks with dimensions 4–6, its performance and sample efficiency on higher-dimensional tasks are not discussed.

---

> ### Author Rebuttal · Authors · 2026-03-31
>
> We thank the reviewer for the constructive feedback. We address each point below.
>
> > **[W1 & Q2] Dependence on LLM capability, GP accuracy, and performance in low-resource environments.**
>
> **PiEvo intentionally decouples complex reasoning from the LLM.** By delegating principle evolution to a structured Bayesian mechanism, the LLM is restricted to constrained template generation rather than deep chain-of-thought inference.
>
> * **Low-resource efficacy.** Activating complex internal reasoning empirically degrades PiEvo's performance by 26.35% (Appendix E). To validate low-resource robustness, we evaluated PiEvo using the lightweight Qwen3-14B (**Table R1**). It maintains strong performance and significantly outperforms baselines, proving it does not rely on computationally expensive reasoning models.
>
> **Table R1. Performance comparison using the lightweight Qwen3-14B backbone.**
> | Method | SQ %| AUOC %| APD %| Performance Drop % |
> | :--- | :--- | :--- | :--- | :--- |
> | PiEvo (ours) | 130.01 $\pm$ 52.60 | 108.3 $\pm$ 40.9 | 24.5 $\pm$ 13.7 | - |
> | PiFlow | 59.99 $\pm$ 9.84 | 50.3 $\pm$ 10.7 | 9.7 $\pm$ 6.2 | - 53.86% |
> | AI-Researcher | 49.42 $\pm$ 15.45 | 40.3 $\pm$ 2.9 | 3.3 $\pm$ 3.0 | - 61.99% |
> | The-AI-Scientist-v1 | 67.09 $\pm$ 13.34 | 60.7 $\pm$ 9.8 | 4.4 $\pm$ 4.8 | - 48.40% |
> | The-AI-Scientist-v2 | 61.87 $\pm$ 4.75 | 61.2 $\pm$ 5.6 | 2.4 $\pm$ 1.5 | - 52.41% |
>
> * **GP predictive accuracy.** Geometric alignment is a robust proxy for experimental likelihood ($R^2 = 0.823$, refer to Figure 12, Appendix G). GP experts associated with valid principles consistently achieve high accuracy across all tasks (refer to **Table R2**). **Crucially, a low $R^2$ is an intended epistemic signal, not a failure.** Per Propositions A.9 and B.1, poor GP fit triggers anomaly-based evolution, allowing PiEvo to falsify incorrect scientific priors rather than over-optimizing them.
>
> **Table R2. Statistical analysis of GP predictive accuracy across all tasks.**
> | Task | Mean and std of $R^2$ | Proportion of Valid Principles ($R^2$ > 0.1) |
> | --- | --- | --- |
> | MBO | 0.9439 $\pm$ 0.0630 | 80.0% |
> | NHO | 0.8115 $\pm$ 0.1345 | 42.9% |
> | SPO | 0.8736 $\pm$ 0.0837 | 95.0% |
> | TMC | 0.8095 $\pm$ 0.1341 | 93.9% |
>
> > **[Q1] Sensitivity of anomaly thresholds and automatic tuning mechanisms.**
>
> Per Eq. 3, the Surprisal Score normalizes deviation $(y_s - \mu_{MAP}(h_s))^2$ by the predictive and observation variance $(\sigma_{MAP}^2 + \sigma_{obs}^2)$, adapting intrinsically to varying domain scales and experimental noise. Consequently, the threshold $\theta_t$ achieves theoretical cross-domain scale invariance. This adaptive normalization explains PiEvo's stable performance across diverse $\theta_t$ values (refer to Figure 5 ablation), eliminating the need for domain-specific manual tuning.
>
> > **[W2] Gap between hierarchical theory and marginalized IDS in practice.**
>
> We address this algorithmic nuance in Appendix A.
> * **Theoretical equivalency.** While our regret decoupling proof assumes hierarchical sampling (sampling $P_t$, then $h_t$), the practical algorithm employs marginalized Information-Directed Sampling (IDS), selecting $h_t$ by marginalizing over the principle posterior $p_t(P)$. This adaptation perfectly preserves theoretical guarantees because it directly minimizes $\Delta_t^2 / I_t(P;Y_t)$, the exact core objective dictating cumulative identification regret $\sum \Delta_t^{(ID)}$ in our proof.
> * **Empirical benefits.** Acting as a robust estimator for expected information gain, marginalized IDS maintains the sublinear $O(\sqrt{T})$ convergence rate (Figure 6). Furthermore, posterior marginalization smooths principle-space exploration, effectively preventing premature convergence to false initial priors during periods of high early-stage epistemic uncertainty (Watershed Phenomenon, Appendix F, Figure 10a).
>
> > **[W3 & Q3] Scalability, sample efficiency, and convergence in higher dimensions.**
>
> **Clarification.** The main NHO benchmark (refer to Table 1) evaluates the **full 7-dimensional space** (structure, wavelength, light direction, observation axis), where PiEvo significantly outperforms baselines ($SQ=96.36\%$). Table 3 limits dimensions solely for ablation purposes.
>
> Regarding scaling, the current lack of high-fidelity, multi-dimensional public scientific surrogate models restricts further empirical testing. However, in **Appendix C**, we prove PiEvo's learning complexity is bounded by $O(\sqrt{T \cdot H(\mathcal{P})})$, where $H(\mathcal{P})$ is the principle space entropy. While traditional hypothesis search agents scale with hypothesis space entropy $H(\mathcal{H})$, which explodes exponentially in higher dimensions (refer to Figure 6), PiEvo remains highly efficient because human-readable scientific principles remain compact and abstract regardless of underlying dimensionality.
>
> We hope these clarifications address your concerns and thank you for helping us improve our manuscript.

---

> > ### Author Rebuttal · Reviewer_7yvo · 2026-04-01
> >
> > The author has addressed my concerns quite well. Although I am not an expert in this specific field, the author has demonstrated the ability to maintain good performance even in low-resource scenarios, which I find impressive. After reviewing the feedback from the other reviewers, I believe a score of 4 may be appropriate.

---

### Official Review · Reviewer_uxSf · 2026-03-11

**Soundness:** 3
**Presentation:** 3
**Significance:** 3
**Originality:** 3
**Overall Recommendation:** 4
**Confidence:** 4

**Summary:**

The paper describes a principle-evolvable framework, PIEVO, to expand possible scientific principles for acceleration of scientific discovery by anomaly-driven prompting for hypothesizing given experimental outcomes. PIEVO serves as a strategic layer to coordinate corresponding large language model (LLM) agents to address the problem of a static hypothesis space adopted in existing multi-agent AI systems for scientific discovery, causing hypothesizing inefficiency by fixed initial priors and the restriction of the discovery of novel phenomena. The authors provided both theoretical analyses and empirical results to support the efficiency of PIEVO.

**Compliance With Llm Reviewing Policy:**

Affirmed.

**Final Justification:**

The authors provided clarifications for the raised questions and concerns with explanations on the adoption of IDS in PiEVO.

**Key Questions For Authors:**

- The authors excluded baselines that are tailored to specific domains. If PIEVO is intended to be a general discovery framework, why was it not evaluated on those domain-specific tasks? Evaluating PIEVO on such tasks could further demonstrate its general applicability and robustness.
- The authors are excluding approaches based on hypothesis-testing mechanisms, stating that these paradigms are already represented by the selected baselines. Can authors clarify why instead of evaluating these approaches, they were completely ignored? Although similar, these methods can provide a more comprehensive comparison.
- Why all approaches are not tested with the Internet access? This can better illustrate the real-world applications of agents

**Limitations:**

The authors may want comment on potential hallucination risks by LLMs and other potential limitations relying on general-purpose LLM agents for scientific discovery.

**Strengths And Weaknesses:**

### Strengths

- PIEVO allows agents to adapt to new data and discover novel phenomena, and avoid epistemic stagnation, by introducing anomaly-driven principle evolution as additional degree of freedom for AI augmented scientific discovery.
- PIEVO models the likelihood of experimental outcomes using Gaussian Process experts for efficient computation. By identifying and reconciling anomalies by proposing new principles, PIEVO ensures that the agent’s internal model scales with empirical data, from computational simulations and experiments ultimately.
- PIEVO was tested on four scientific discovery benchmarks: nanomaterial optical property optimization (NHO), molecular bio-activity optimization (MBO), superconductor critical temperature optimization (SPO), and transition metal complex optimization (TMC) and reported consistent performance improvements over existing multi-agent systems.

### Weaknesses

- PIEVO appears to use PiFlow as the baseline, with added PIEVO strategic layer. Considering that scientific principle evolution is still based on LLM prompting, the authors may articulate how performances PIEVO will change if considering principle prompts and hypothesis prompts from the corresponding Principle Agent and Hypothesis Agent are treated as more complex prompt structures, such as chain-of-thoughts, using one Hypothesis Agent instead.
- The presented theoretical analysis is rather decorative with strong assumptions on LLM generated principles and hypotheses, especially challenging to justify considering the lack of prompt optimization guarantees considering LLM outputs.
- For the reported empirical studies, the authors may consider comparing with domain specific approaches besides multi-agent systems to really establish that PIEVO can accelerate the corresponding scientific discovery. Otherwise, it is just another potential multi-agent system framework as all the existing ones.
- No code was provided for reproducibility checking.

---

> ### Author Rebuttal · Authors · 2026-03-31
>
> We thank the reviewer for this insightful feedback. Below we address the concerns:
>
> > **[W1] On merging principle and hypothesis agents into a single chain-of-thought structure**
> >
>
> Treating the *Principle* as a structurally separate entity (rather than a single agent for both principle and hypothesis) is mandatory for PiEvo: **The principle must be isolated as a discrete semantic constraint to train Gaussian Process** and dynamically compute the Bayesian posterior belief $p_t(P)$. Single complex prompt, thereby treating principle as a part of Chain-of-Thought, cannot yield a quantifiable epistemic state.
>
> Regarding the combination of Chain-of-Thought on PiEvo architecture and what will happen in this setting: **we explored this exact dynamic in Appendix E, Table 7.** We compare PiEvo against AI Researcher using highly complex internal reasoning structures (Gemini-2.5-Flash Think mode, which acts as a sophisticated CoT).
>
> We found that while AI Researcher slightly benefit from CoT prompts (SQ improved by 25.06%), PiEvo's performance strictly *degraded* by 26.35% when applying complex reasoning in a single prompt. We hypothesize that this occurs due to **cognitive interference**, as detailed in Appendix E.
>
> > **[W2] The theory seems decorative with strong assumptions on LLM outputs without prompt optimization guarantees.**
> >
>
> **We respectfully clarify that our theory is foundational to the system's architecture, not decorative.** The Information-Directed Sampling formula (Equation 1) in PiEvo, is mathematically derived directly from the theoretical regret bound established in **Theorem 3.3.**
>
> Regarding the LLM assumptions, **our theory does *not* assume perfect prompts or flawless LLM outputs.** In **Appendix A.1 (Assumptions G1-G3)**, we explicitly model the LLM purely as a noisy, finite-entropy generator. The theoretical framework handles the imperfect nature of LLMs by strictly decoupling the **Identification Cost** (finding the right physical principle) from the **Hypothesis Generation Cost**. We mathematically prove that even if the LLM's prompt generates suboptimal candidates, bounding its decoding entropy keeps the systemic regret sublinear.
>
> > **[W3, Q1, Q2] Why exclude specific approaches (Co-scientist, Kosmos)? Why not compare PiEvo with domain-specific approaches to establish true acceleration?**
> >
>
> We excluded domain-specific systems to prevent confounding variables (e.g., specialized knowledge graphs, retrieval pipelines) and isolate PiEvo's intrinsic reasoning efficacy. However, to address your question, we evaluated PiEvo against standard domain-agnostic optimizers. As shown below, PiEvo significantly outperforms traditional baselines.
>
> **Table R1: Solution Quality (SQ %) under the same 24-step budget**
> | Method | MBO | NHO | SPO | TMC | Average |
> | --- | --- | --- | --- | --- | --- |
> | **Random Search** | 62.7 ± 4.1 | 39.5 ± 16.3 | 35.8 ± 1.2 | 70.0 ± 5.4 | 52.0 |
> | **Genetic Algorithm (GA)** | 65.9 ± 7.2 | 44.8 ± 19.3 | 39.4 ± 2.6 | 74.2 ± 6.3 | 56.1 |
> | **Bayesian Optimization (BO)** | 70.2 ± 9.5 | 64.8 ± 12.2 | **42.6 ± 1.0** | 76.6 ± 4.7 | 63.5 |
> | **Differential Evolution (DE)** | 62.7 ± 4.1 | 51.2 ± 14.3 | 37.2 ± 1.6 | 70.0 ± 5.4 | 55.3 |
> | **PiEvo w/ Qwen** | 149.1 ± 16.0 | **96.4 ± 1.7** | 37.3 ± 1.4 | 80.5 ± 1.9 | 90.8 |
> | **PiEvo w/ Gemini** | **153.5 ± 6.4** | 88.0 ± 1.1 | 37.9 ± 0.4 | **93.3 ± 0.8** | **93.2** |
>
> **Traditional solvers (BO, DE) struggle in few-shot regimes because they build response surfaces from scratch.** Conversely, **PiEvo acts as a powerful zero-shot prior generator, mitigating the cold-start problem.** As demonstrated in our FDTD-validated Nanohelix study (refer to Section 4.5), PiEvo's evolved principles structurally constrain the search space to high-yield manifolds without external retrieval. We will gladly include these baselines and discussions in the final manuscript.
>
> > **[Q3] Why are all approaches not tested with Internet access?**
> >
>
> **Web access was disabled to evaluate genuine *scientific discovery* over mere information retrieval.** In the SPO task, internet access allows trivial lookup of the highest $T_c$ superconductor, bypassing the scientific method entirely. Restricting access forces agents to genuinely deduce mechanisms and evolve principles from raw experimental feedback.
>
> > **[L1] What about hallucination risks relying on general-purpose LLMs?**
> >
>
> **PiEvo mathematically neutralizes hallucinations via GP experts and anomaly detection.** If an LLM hallucinates a physically incorrect principle, its corresponding GP expert will yield massive predictive errors against surrogate data. This generates a high anomaly score (Eq. 3), driving the hallucinated principle's Bayesian posterior $p_t(P)$ to zero and actively purging it from memory.
>
> We will add a Limitations subsection to the discussion making this mechanism explicit, and will release all code, prompts, and surrogate models for full reproducibility upon acceptance.

---

> > ### Author Rebuttal · Reviewer_uxSf · 2026-04-02
> >
> > Thanks for the rebuttal. The authors may want to help clarify how the results on regret bounds in Theorem 3.3 (Appendix) led to the adopted IDS.

---

> > > ### Author Response · Authors · 2026-04-03
> > >
> > > We sincerely thank the reviewer for pointing this out. We recognize that our use of the word *"derived"* in the previous context may have caused confusion. We wish to clarify that **our adoption of IDS was not an arbitrary heuristic choice, but rather a necessary algorithmic consequence** deduced from our structural decomposition of the scientific discovery problem. Here is our design rationale:
> > >
> > > We formulated scientific discovery as a dual-uncertainty problem. By decomposing the instantaneous regret $\Delta_t$, we identified that the **primary theoretical bottleneck** for achieving convergence is bounding the cumulative **identification regret**, $\sum_{t=1}^T \Delta_t^{(ID)}$.
> > >
> > > How do we bound the cost of identifying the right principle? **We recognized that the universal principle space has a fundamental property: a finite prior entropy $H(p_0)$**. Every experimental outcome provides some information gain $I_t$ about the true principle, and the cumulative information gain is strictly bottlenecked by this finite prior entropy: $\sum_{t=1}^T I_t \le H(p_0)$
> > >
> > > **To guarantee that the identification regret stops growing**, we must strictly anchor the accumulation of regret to this finite information budget. In other words, we cannot afford exploratory actions that incur high regret but yield zero information. Therefore, we can algebraically construct the connection between the cumulative identification regret and the information gain by writing:
> > > $$\sum_{t=1}^T \Delta_t^{(ID)} = \sum_{t=1}^T \left( \frac{\Delta_t^{(ID)}}{\sqrt{I_t}} \cdot \sqrt{I_t} \right)$$
> > >
> > > By the Cauchy-Schwarz inequality, we have:
> > > $$\sum_{t=1}^T \Delta_t^{(ID)} \le \sqrt{ \sum_{t=1}^T \frac{(\Delta_t^{(ID)})^2}{I_t} } \cdot \sqrt{ \sum_{t=1}^T I_t }$$
> > >
> > > As established, the total information gain is bounded:
> > > $$\sum_{t=1}^T I_t \le H(p_0)$$
> > >
> > > Therefore, the inequality becomes:
> > > $$\sum_{t=1}^T \Delta_t^{(ID)} \le \sqrt{ \sum_{t=1}^T \frac{(\Delta_t^{(ID)})^2}{I_t} } \cdot \sqrt{ H(p_0) }$$
> > >
> > > **If the system ensures that $\frac{(\Delta_t^{(ID)})^2}{I_t} \le \Gamma$ at each step**, then $\sum_{t=1}^T \frac{(\Delta_t^{(ID)})^2}{I_t} \le \Gamma T$. Substituting this into the equation above, we obtain our sublinear bound:
> > > $$\sum_{t=1}^T \Delta_t^{(ID)} \le \sqrt{\Gamma T} \cdot \sqrt{H(p_0)} = \mathcal{O}(\sqrt{T})$$
> > >
> > > **This imposes a strict mathematical design constraint**: to guarantee sublinear convergence ($\mathcal{O}(\sqrt{T})$), the system's hypothesis-selection mechanism must explicitly constrain the ratio of squared regret to information gain ($\frac{(\Delta_t^{(ID)})^2}{I_t}$) at each step.
> > >
> > > We sought a strategy that explicitly operationalizes this mathematically required regret-to-information trade-off. **IDS is precisely formulated to minimize this exact ratio.** Therefore, the architecture of PiEvo naturally necessitates the adoption of IDS over standard UCB or greedy generation, because its objective function structurally guarantees the algebraic prerequisite needed to exploit the bounded entropy of the principle space.

---

### Official Review · Reviewer_zvR9 · 2026-03-13

**Soundness:** 3
**Presentation:** 3
**Significance:** 4
**Originality:** 4
**Overall Recommendation:** 5
**Confidence:** 2

**Summary:**

This manuscript presents PIEvo, a framework that reformulates scientific discovery as Bayesian optimisation over an evolving principle space rather than choosing between a set of predetermined hypothesis. The system uses Information-Directed Sampling via Gaussian Process experts and an anomaly-driven augmentation mechanism. Evaluation across four benchmarks (NHO, MBO, SPO, TMC) demonstrates substantial performance gains over existing methods.

**Compliance With Llm Reviewing Policy:**

Affirmed.

**Final Justification:**

Thanks for the rebuttal. I remain confident that this is a good paper and keep my scores

**Key Questions For Authors:**

Could you provide a concrete example of a principle P and several hypotheses h₁, h₂, h₃ generated under that principle, showing the actual text/structure? This would greatly aid understanding.

Line 160: Please clarify explicitly how f* maps hypotheses to outcomes, again giving a worked example.

Please can you elaborate on the GP framework (see comment above).

**Limitations:**

Yes

**Strengths And Weaknesses:**

# Strengths
1. Novel conceptual contribution: Overall, an important concept considered by this manuscript is the shift from searching within a fixed hypothesis space to optimising over an evolvable principle space. This is a genuinely interesting departure from prior work and addresses a real limitation in current LLM-based scientific agents.

2. Strong empirical results: The performance improvements are impressive

3.  Theoretical grounding: The authors provide rigorous theoretical analysis (Appendix A), including regret bounds and posterior concentration guarantees. Theorem A.6 and the subsequent propositions offer valuable formal justification for the approach. Note that I am not an expert in regret bounds and so cannot comment on its validity

4. Information-directed sampling: Section 3.3 feels like the correct thing to do in this setting, and seems more principled than the other compared algorithms (although I am not an expert). The IDS formulation (Equation 1) appropriately balances regret against information gain.

# Weaknesses

I am not an expert in this specific area (principle-evolvable scientific discovery), which limits my confidence. My primary concerns are:

1. Clarity of fundamental setup: I found it very hard to understand the problem setting (I had to read referenced papers). The authors attempt to focus on a broad topic but assume readers already understand what constitutes a "principle" versus a "hypothesis" in their framework. A concrete, worked example early in the paper would substantially improve accessibility(perhaps in Section 3.1), e.g.  showing: (i)What a principle looks like (with actual text), (ii) What its associated hypotheses look like, and (iii) How a function maps a hypothesis to a single value y (line 160). I expect ICML papers to be more self-contained, especially when introducing novel problem formulations.


2. I would like more clarity on the use of GPs.  The use of GP experts to model likelihoods p(y|h,P) via semantic embeddings (Equation 2) is intriguing. I can't find details on the choice of kernel and/or info on how the kernel parameters are fit. I also worry about the suitability of the kernel choice. Significant work has gone into designing specific GPs for high-dim problems (see refs below) and it would be interesting to see if any of the advances improve things.

3. There is limited discussion of failure modes: When does PIEvo fail? For example, what happens when: (i)The true principle genuinely lies outside the LLM's knowledge base?, (ii) Multiple principles explain the data equally well?, or (iii) The GP experts fail to maintain calibrated uncertainty?


Hvarfner, Carl, Erik O. Hellsten, and Luigi Nardi. "Vanilla Bayesian optimization performs great in high dimensions." Proceedings of the 41st International Conference on Machine Learning. 2024.

Duvenaud, David K., Hannes Nickisch, and Carl Rasmussen. "Additive gaussian processes." Advances in neural information processing systems 24 (2011).

Lu, Xiaoyu, Alexis Boukouvalas, and James Hensman. "Additive Gaussian processes revisited." International conference on machine learning. PMLR, 2022.

---

> ### Author Rebuttal · Authors · 2026-03-31
>
> We thank the reviewer for the highly positive evaluation and insightful feedback. Below, we address the questions and outline the improvements we will make in the final version.
>
> > **[W1, Q1, Q2]  Clarity of fundamental setup and an example.**
> >
>
> We agree that a concrete, up-front example would significantly improve readability. As you suggested, we will add a *Running Example* box in Section 3.1. To answer your questions directly, here is the concrete mapping from our Nanohelix Optimization (NHO) task, drawn from Appendix H and I.
>
> - **A principle $P$.** A natural language physical mechanism proposed by the agent, e.g., "*Maximum g-factor arises from Toroidal Anapole Resonance, which requires a tight helix radius, high turn density, and a short pitch length to enable destructive multipolar interference."*
> - **A hypothesis $h$.** A testable parameter set derived from the premises of $P$, e.g., fiber_radius=20.0, helix_radius=60.0, n_turns=6.0, pitch=20.0, wavelength=400.0, x_y_z=2, direction=0.
> - **External validation function $f^*$.** The true underlying physical mapping, represented in our benchmarks by a high-fidelity surrogate model (Section 3.2). It takes the concrete hypothesis $h$ as input and outputs the scalar observation $y$.
>
> > **[W2, Q3] GP framework details and the curse of dimensionality**
> >
>
> We appreciate the references to Additive GPs for high-dimensional Bayesian Optimization. However, PiEvo  bypasses the high-dimensional BO curse entirely via a semantic bottleneck design.
>
> - **Theoretical design.** Our GP experts do not operate directly on the high-dimensional hypothesis space. As defined in Eq. 2 and proven in Appendix B, the GP input is $\phi(h, P)$, a highly compressed 2D semantic alignment feature vector consisting solely of the cosine similarity $\langle e_h, e_P \rangle$ and Euclidean distance $||e_h - e_P||_2$ between the hypothesis and principle embeddings.
> - **Empirical validation for kernel choice.** Because the space is strictly 2-dimensional, standard kernels are both sufficient and highly effective. To empirically address your suggestion, we evaluated an Additive Kernel against our standard RBF kernel on this 2D space:
>
>
>     | Method | SQ (%) | AUOC | APD |
>     | --- | --- | --- | --- |
>     | RBF Kernel (2D) | 149.06±13.08 | 1.183±0.194 | 0.245±0.021 |
>     | Additive Kernel (2D) | 161.46±61.78 | 1.142±0.320 | 0.213±0.060 |
>
>     While the Additive Kernel shows a slightly higher mean SQ, it introduces severe instability (massive variance) and degrades exploitation/exploration balance (lower AUOC/APD). Thus, the RBF kernel (Algorithm 1) remains the optimal choice for our low-dimensional feature space. Furthermore, GP hyperparameters are dynamically fit via marginal log-likelihood maximization at each iteration (detailed in Appendix G, Fig 12a). We will explicitly clarify this 2D dimensionality reduction in the main text to prevent future misunderstandings.
>
>
> > **[W3] Discussion of failure modes.**
> >
>
> We appreciate the edge cases raised by the reviewer. We will add a dedicated Discussion on Failure Modes paragraph addressing these exact scenarios:
>
> - **The true principle lies strictly outside the LLM's knowledge base.** This is the boundary of LLM-based agents as a hypothesizer or reasoning machine. While the anomaly-driven augmentation forces the LLM to synthesize new principles to resolve contradictions, if the foundational scientific concepts are entirely absent from LLM's piror, the agent will repeatedly propose flawed principles. The system will stall, reflecting persistent high anomaly scores and an inability to reduce regret.
> - **Multiple principles explain the data equally well.** This is elegantly handled by Information-Directed Sampling in PiEvo. If two principles have identical predictive likelihoods for past data, the posterior belief is split. IDS (Eq. 1) will actively seek a new hypothesis $h_t$ that yields the highest *Information Gain,* specifically looking for an experimental parameter where the two principles predict *divergent* outcomes. If no such hypothesis exists in the entire universe, the principles are mathematically/physically equivalent, and optimizing either yields the same optimum.
> - **The GP experts fail to maintain calibrated uncertainty.** If GP calibration fails by misestimating variance $\sigma^2$, the anomaly score $S_s$ (Eq. 3) becomes unreliable. The system might suffer from hallucinated anomalies (needlessly expanding the principle space) or missed anomalies (premature convergence on a flawed principle). As shown in Figure 5, our sensitivity analysis on the noise scale and threshold shows the system is robust within reasonable bounds, but catastrophic miscalibration would break the evolutionary loop.
>
> We hope these clarifications and our commitment to integrating the concrete examples and failure mode discussions directly address your concerns.

---

> > ### Author Rebuttal · Reviewer_zvR9 · 2026-04-01
> >
> > Thanks for the rebuttal. I maintain my high score

---

### Official Review · Reviewer_FHjk · 2026-03-14

**Soundness:** 3
**Presentation:** 3
**Significance:** 3
**Originality:** 3
**Overall Recommendation:** 5
**Confidence:** 3

**Summary:**

This paper proposes PIEVO, a scientific discovery framework that evolves principles rather than searching hypotheses under fixed principles. The method combines principle posterior updating, information-directed hypothesis selection, and anomaly-triggered principle augmentation. Results on multiple science benchmarks are strong and consistently better than prior agent baselines.

**Compliance With Llm Reviewing Policy:**

Affirmed.

**Key Questions For Authors:**

1) How should “principles” be interpreted in this framework: as scientific theories, semantic priors, or structured natural-language heuristics?

2) How robust is the anomaly-triggered augmentation step when the observed anomaly is caused by noise or surrogate model error rather than genuine principle mismatch?

3)Can the authors clarify the distinction between PIEVO and evolutionary code-discovery frameworks such as AlphaEvolve, especially in terms of scope, representation, and applicability?

**Limitations:**

The notion of “principle” is still under-formalized, and in practice appears closer to evolving semantic priors than fully specified scientific theories.

**Strengths And Weaknesses:**

Strengths: The paper has a clear and novel high-level idea: letting the system revise its guiding principles instead of treating them as fixed. The framework is coherent, technically well organized, and the empirical gains are strong. This is a meaningful step beyond standard agentic search.


Weakness: The notion of “principle” is still not fully formalized; in practice it looks closer to evolving semantic priors than fully specified scientific theories. Also, the paper would benefit from clearer positioning against code-evolution frameworks such as AlphaEvolve.

---

> ### Author Rebuttal · Authors · 2026-03-31
>
> We thank the reviewer for recognizing PiEvo’s novelty, technical organization, and strong empirical performance.
>
> > **[W1 & Q1] Interpreting “Principles”**
> >
>
> We agree: in PiEvo, principles function as evolvable semantic priors, not formal axiomatized theories. We will clarify this in the revised Intro and Section 3.
>
> - **Formally,** they are natural-language heuristics structuring LLM reasoning without imposing symbolic rigidity, refer to Section 3.1–3.2.
> - **Empirically,** despite their linguistic form, these priors capture complex physics (e.g., successfully identifying the Toroidal anapole resonance in nanophotonics through iterative refinement).
>
> We will add a dedicated discussion noting how LLM hallucinations or pretraining biases can temporarily mislead search, identifying hybrid/symbolic representations as a vital direction for future work.
>
> > **[Q2] Robustness of the anomaly-triggered augmentation**
> >
>
> PiEvo distinguishes genuine mismatch from surrogate error via GP uncertainty gating.
>
> - **Case study: a successfully rejected false anomaly.** To concretely illustrate this robustness, we extracted a specific trajectory from our MBO task experiments where PiEvo successfully intercepted a *false anomaly* caused by surrogate inaccuracy.
>     - At iteration 13 (24 iterations in total), under the MAP principle of `aromatic interaction optimization`, a candidate molecule exhibited a severe prediction deviation, i.e., with prediction of 4.553 but the actual value is 2.588. A naive error-based trigger would have immediately forced a spurious principle augmentation.
>     - **However, because the GP model had not yet gathered sufficient data in this specific chemical subspace, its predictive variance ($\sigma_{MAP}^2 \approx 1.1$) remained notably high.** Per Eq. 3, this large epistemic uncertainty dominated the denominator, effectively damping the Anomaly Score $S_s$ and keeping it below the activation threshold ($\theta_t = 0.85$).
>     - Consequently, the system correctly classified the deviation as data insufficiency rather than principle failure, and deferred augmentation. By round 15, after sufficient observations had reduced $\sigma_{MAP}^2$, persistent deviations were accurately verified as genuine mismatches, triggering a well-founded principle refinement. GP uncertainty thus acts as a strict epistemic gate that prevents premature augmentation.
> - **Theoretically, the anomaly score $S_s$ does not rely on raw prediction error.**
>     - As formalized in Equation 3, the metric is normalized by the sum of the GP expert's predictive variance ($\sigma_{MAP}^2(h_s)$) and the explicit observation noise variance ($\sigma_{obs}^2$, as shown in Figure 5). Mathematically, an anomaly is only flagged when the empirical deviation is statistically significant relative to both the known *aleatoric noise* and the model's current *epistemic uncertainty*.
> - **Empirically, we evaluated PiEvo's resilience to noise in Section 4.6.**
>     - As demonstrated in Section 4.6, Figure 5, PiEvo maintains robust performance across a spectrum of anomaly thresholds ($\theta_t$) and noise scales. Furthermore, the GP experts dynamically adapt their hyperparameters (Figure 12a) to calibrate against task-specific noise patterns, effectively preventing the system from indiscriminately augmenting the principle space due to routine surrogate fluctuations.
>
> > **[W2, Q3] Distinction from evolutionary code-discovery frameworks**
> >
>
> PiEvo and code-evolution frameworks such as AlphaEvolve both use LLMs within an evolutionary loop, but they differ on three dimensions that determine where each method applies:
>
> - **Search representation.**
>     - Code-evolution frameworks like *AlphaEvolve* search over **procedural** parameters, representing solutions as executable code or programmatic logic.
>     - PiEvo represents solutions structurally, applying Bayesian updates to **semantic** principles expressed in natural language that encapsulate physical or chemical phenomena.
> - **Optimization mechanism.**
>     - Code-evolution generally operates via **mutation**/crossover heuristics or open-ended programmatic refinement.
>     - In contrast, PiEvo is grounded in a **Bayesian optimization** framework for optimizing the strategy, to better make scientific hypotheses.
> - **Target domain and feedback regime.**
>     - Code-evolution is primarily applied to algorithmic tasks where environmental feedback is **deterministic** and computational budgets are vast.
>     - PiEvo is specifically tailored for the natural sciences (e.g., molecular bio-activity, nanophotonics). In these domains, experimental feedback is **stochastic and budgets are severely restricted**, demanding the extreme sample efficiency that PiEvo's semantic abstraction provides.
>
> We will add a dedicated paragraph in the revised manuscript contrasting PiEvo with code-evolution frameworks along these three dimensions.

---

> > ### Author Rebuttal · Reviewer_FHjk · 2026-04-04
> >
> > Thank you for the detailed rebuttal.  My main concerns have been adequately addressed, and I believe my original score remains appropriate.

---

### Decision · Program_Chairs · 2026-04-30

**Decision:**

Accept (regular)

**Comment:**

This paper presents a framework that reformulates scientific discovery as Bayesian optimization over an evolving principle space rather than choosing between a set of predetermined hypotheses. The proposed method, PiEvo, integrates Information-Directed Hypothesis Selection via Gaussian Process and an anomaly-driven augmentation mechanism, which enables agents to autonomously refine their theoretical worldview. The experimental results on benchmarks demonstrate the superiority of PiEvo compared to prior agent baselines.

The concept of this paper is novel and important, enabling the expansion of the principles. The proposed framework is technically well-organized. The empirical results clearly show the advantage of PiEvo.

As the reviewer pointed out, the term "principle" should be further formalized in the revised paper, including a concrete example.

All reviewers recognize the novelty and effectiveness of the paper. Therefore, I would recommend accepting this paper.